



# A Multiphase CMAQ Version 5.0 Adjoint

Shunliu Zhao[1], Matthew G. Russell[1], Amir Hakami[1], Shannon L. Capps[2], Matthew D. Turner[3], Daven K. Henze[4], Peter B. Percell[5], Jaroslav Resler[6], Huizhong Shen[7], Armistead G. Russell[7], Athanasios Nenes[8,9,10], Amanda J. Pappin[11], Sergey L. Napelenok[12], Jesse O. Bash[12], Kathleen M. Fahey[12], Gregory R. Carmichael[13], Charles O. Stanier[13], Tianfeng Chai[14]

[1]Department of Civil and Environmental Engineering, Carleton University, Ottawa, ON K1S 5B6, Canada
[2]Civil, Architectural, and Environmental Engineering, Drexel University, Philadelphia, PA 19104, USA
[3]SAIC, Stennis Space Center, MS 39529, USA
[4]Mechanical Engineering Department, University of Colorado, Boulder, CO 80309, USA
[5]Department of Earth & Atmospheric Sciences, University of Houston, Houston, TX 77204, USA
[6]Institute of Computer Science of the Czech Academy of Sciences, Prague, 182 07, Czech Republic
[7]School of Civil and Environmental Engineering, Georgia Institute of Technology, Atlanta, GA 30331, USA
[8]School of Earth and Atmospheric Sciences, Georgia Institute of Technology, Atlanta, GA 30331, USA
[9]School of Architecture, Civil & Environmental Engineering, Ecole polytechnique fédérale de Lausanne, CH-1015, Lausanne, Switzerland
[10]Institute for Chemical Engineering Sciences, Foundation for Research and Technology Hellas, Patras, GR-26504, Greece
[11]Air Health Effects Division, Health Canada, Ottawa, ON K1A 0K9, Canada
[12]Atmospheric & Environmental Systems Modeling Division, U.S. EPA, Research Triangle Park, NC 27711, USA
[13]Department of Chemical and Biochemical Engineering, University of Iowa, Iowa City, IA 52242, USA
[14]College of Computer, Mathematical, and Natural Sciences, University of Maryland, College Park, MD 20742, USA

*Correspondence to*: Amir Hakami (amir.hakami@carleton.ca)

**Abstract.** We present the development of a multiphase adjoint for the Community Multiscale Air Quality (CMAQ) model, a widely used chemical transport model. The adjoint provides location- and time-specific gradients that can be used in various applications such as backward sensitivity analysis, source attribution, optimal pollution control, data assimilation and inverse modeling. The science processes of the CMAQ model include gas-phase chemistry, aerosols, cloud chemistry and dynamics, diffusion and advection. Discrete adjoints are implemented for all the science processes, with an additional continuous adjoint for advection. The development of discrete adjoints is assisted with Algorithmic Differentiation (AD) tools. Particularly, the Kinetic PreProcessor (KPP) is implemented for gas-phase and aqueous chemistry, and two different automatic differentiation tools are used for other processes such as clouds, aerosols, diffusion, and advection. The continuous adjoint of advection is developed manually. For adjoint validation, the brute-force or Finite Difference Method (FDM) is implemented process by process with box- or column-model simulations. Due to the inherent limitations of the FDM caused by numerical round-off errors, the Complex Variable Method (CVM) is adopted where necessary. The adjoint model often shows better agreement with the CVM than with the FDM. The adjoints of all science processes compare favorably with the FDM/CVM. In an example application of the full, multiphase adjoint model, we provide the first estimates of how emissions of $PM_{2.5}$ affect public health across the US.



## 1 Introduction

Adjoint models generate gradients which can be used directly for backward sensitivity analysis or to provide directions for gradient-based optimization in four-dimensional variational data assimilation (4D-Var) or other inverse problems (Errico, 1997; Navon, 1997; Giles and Pierce, 2000; Wang et al., 2001; Sandu et al., 2005; Griewank, 2012). Applications of adjoint

models for data assimilation have a long successful history in meteorology and oceanography (Errico, 1997; Navon, 1997). For atmospheric chemistry, adjoint modeling was used as early as the 1990s (Fisher and Lary, 1995; Elbern et al., 1997). More recently the methods were applied to aerosols in 1D models (Henze et al., 2004; Sandu et al., 2005), 3D models of inert aerosol mass concentrations (Hakami et al., 2005), and 3D models of chemically active aerosol mass concentrations (Henze et al., 2007).

Due to omnipresent uncertainties in emissions, initial and boundary conditions, as well as the underlying complex physical and chemical processes, predicting or accurately simulating air quality poses great challenges; assimilation of chemical data is thus a promising approach in improving the model skill (Carmichael et al., 2008). While applications in data assimilation and inverse modeling form the traditional niche for adjoint applications in atmospheric modeling, adjoint models can also be

used to conduct sensitivity analysis. Sensitivity analyses are often performed in air quality studies to estimate the impact of various model inputs, in particular emissions, on model predictions (Dunker et al., 2002; Hakami et al., 2003; Sandu et al., 2005; Cohan et al., 2005; Napelenok et al., 2006; Martien and Harley, 2006; Koo et al., 2007). Among various methods for sensitivity analysis, two general categories are more commonly used: forward and adjoint (Hakami et al., 2007). In the forward approach, sensitivity information is propagated forward in time. The most common forward sensitivity approach is the brute-

force or Finite Difference Method (FDM). The FDM requires minimal effort to implement, but the search for a proper step/perturbation size might be needed to produce accurate sensitivities (Iott et al., 1985). The step size selection process could be resource demanding and repeatedly required, especially when the numerical model contains highly nonlinear processes that are routinely encountered in atmospheric models. Another forward approach is the Complex Variable Method (CVM) (Squire and Trapp, 1998; Anderson and Nielsen, 2001). Unlike the FDM, the CVM is not subject to subtraction errors and can provide

accurate sensitivities by using a perturbation size as small as allowed in floating-point calculations, but this approach has only been implemented in one atmospheric chemical transport model (Giles et al., 2003; Constantin and Barrett, 2014). A third forward approach is the Decoupled Direct Method (DDM) or the Tangent Linear Model (TLM), in which differentiation is directly applied to the governing equations or algorithmically to the primal computer model. DDM can generate exact sensitivities (i.e., subject to numerical errors, with no perturbations required) at the cost of a significant amount of model

development (Dunker et al., 2002; Napelenok et al., 2006).

In the adjoint approach, the sensitivity information is propagated backward in time and a single model run generates sensitivities of a metric of model outputs with respect to all model inputs (Errico, 1997; Giles and Pierce, 2000). The adjoint





and the forward sensitivity approaches complement each other, in the sense that the forward approach calculates sensitivities of all model outputs to a single model input, combined or individual, in one model run (an extra baseline run required for the FDM), while the adjoint method provides sensitivities of a single model output (individual, or integrated) to all model inputs (Hakami et al., 2007).

In addition to 4D-Var and sensitivity analysis, adjoint models have been implemented for source attribution of air pollutants (Zhang et al., 2015; Qi et al., 2017). Compared to the zero-out method which computes contributions by switching an emission sector on and off, the adjoint approach has the advantage of not changing the chemical environment which could lead to inaccuracies in estimates (Koo et al., 2009). Furthermore, the adjoint approach can be readily extended to include more

emission sectors for investigation with a marginal increase in computational cost; as mentioned earlier, the computational cost of adjoint models is practically independent of the number of input parameters. Source attribution by adjoint models has its own limitation due to the inherent linear assumption in adjoint formulation. Koo et al. (2009) found that the linear assumption held for a 20% emission reduction in general for secondary inorganic aerosols; for secondary organic aerosols, the linearity assumption was valid for up to 100% reductions in anthropogenic emissions. Although these bounds are based on the DDM

of the three-dimensional Comprehensive Air quality Model with extensions (CAMx; ENVIRON, 2005; Koo et al., 2007), they are applicable to the bounds for adjoint models for source attribution, as both the DDM and the adjoint are tangent estimations and based on the same assumption of linearity.

The U.S. Environmental Protection Agency's Community Multiscale Air Quality (CMAQ) model is a regional-to-hemispheric

air quality model, which is widely used due to its community-driven development and state-of-the-art science components (Byun and Schere, 2006; Foley et al., 2010). Limited adjoint versions of CMAQ have been developed before; gas-phase adjoint model was previously developed for CMAQ 4.5.1 (Hakami et. al, 2007) and has been used in several applications related to ozone (Resler et al., 2010; Mesbah et al., 2012; Pappin and Hakami, 2013; Zhao et al., 2013; Pappin et al., 2015; Pappin et al., 2016; Park et al., 2016). Turner et al. (2015a; 2015b) developed and applied the adjoint of the black carbon (BC) aerosol for

CMAQ 4.7.1 but did not include other aerosol species or gas-phase chemistry.

The lack of chemically comprehensive aerosol and cloud processes in the adjoint model has so far prevented applications related to aerosols, which in turn has imposed significant limitation on adjoint-based multi-pollutant studies on topics such as human health and climate. Analogous to ozone, exposure to fine particulate matter ($PM_{2.5}$) poses risks to human health (Brook

et al., 2010; Crouse et al., 2012, 2015; West et al., 2016; Turner et al., 2016; Di et al., 2017; Pinault et al., 2017). Particulate matter also plays a significant role in climate change by influencing the radiative budget of the atmosphere (Tai et al., 2010, 2012; Fiore et al., 2012; Fuzzi et al., 2015). A multiphase adjoint model has been shown to better delineate the influence of model inputs such as emissions on human health (Lee et al., 2015; Koplitz et al., 2016) and climate (Henze et al., 2012; Karydis et al., 2012; Lacey et al., 2017).





Adjoint of air quality models have been developed before, but many of these adjoint models were for the gas-phase chemistry and/or contain a less detailed treatment (lacking microphysics or thermodynamics) of aerosols than that of CMAQ (Elbern et al., 2000; Henze et al., 2004; Sandu et al., 2005; Hakami et al., 2005; Martien and Harley, 2006; Henze et al., 2007; Dubovik
et al., 2008; Huneeus et al., 2009), or were developed for a global model (Henze et al., 2007). This work aims to fill in these gaps by developing a full adjoint for a regional air quality model with detailed, multiphase, size-resolved treatment of aerosols, and to modify the adjoint to reflect more recent science process updates present in CMAQ version 5.0.

## 2 Adjoint Model Development

The CMAQ modeling system solves the atmospheric diffusion equations (ADEs, Byun, 1999; Jacobson, 2005; Seinfeld and
Pandis, 2006)

$$\frac{\partial c_i}{\partial t} = -\mathbf{u} \cdot \nabla c_i + \frac{1}{\rho} \nabla \cdot (\rho \mathbf{K} \nabla c_i) + r_i + e_i + s_i \,, \tag{1}$$

where $c_i$ is the mixing ratio of species $i$, $\mathbf{u}$ is the wind velocity, $\rho$ is the air density, $\mathbf{K}$ is the diffusivity tensor, and $r_i$, $e_i$ and $s_i$ represent the change rates from chemical/thermodynamic transformations, emissions, and the loss process for species $i$, respectively. The first two terms on the right-hand side represent the transport process, namely advection and diffusion. Given
the proper initial conditions and boundary conditions, the CMAQ model simulates the fate of air pollutants and their precursors, emitted to or produced via chemical reactions in the atmosphere.

Integration of the ADE in CMAQ is accomplished through operator splitting, which facilitates the modular structure of the model (Byun and Schere, 2006; McRae et al., 1982). CMAQ includes sub-modules implemented for all the science processes:
• VDIFF for vertical diffusion,
     • HADV for horizontal advection,
     • ZADV for vertical advection,
     • HDIFF for horizontal diffusion,
     • CLDPROC for cloud dynamics and aqueous chemistry,
• CHEM for gas-phase chemistry,
     • AERO for aerosol dynamics and thermodynamics.
In the CMAQ model, the science processes are executed one after another at every synchronization time step that is dictated by the stability criteria for horizontal advection (Byun and Schere, 2006). To guarantee accuracy and to meet stability criteria, internal time steps specific to each process are also employed.

The adjoint equations corresponding to the ADEs can be written as



$$-\frac{\partial \lambda_i}{\partial t} = \nabla \cdot (\mathbf{u}\lambda_i) + \nabla \cdot \left(\rho \mathbf{K} \nabla \frac{\lambda_i}{\rho}\right) + \tilde{r}_i + \varphi_i, \tag{2}$$

where $\lambda_i$ represents the adjoint variable of species $i$; $\tilde{r}_i$ represents the contributions from $r_i$, $e_i$ and $s_i$; and $\varphi_i$ denotes adjoint forcing (Elbern et al., 2000; Sandu et al., 2005; Martien and Harley, 2006; Henze et al., 2007; Hakami et al., 2005, 2007). In the following sub-sections, the adjoint model development techniques and strategies are introduced and the details of the

challenges and treatment for each science module are discussed.

## 2.1 Continuous and discrete adjoints

There are two approaches in developing an adjoint model, the discrete and the continuous (Giles and Pierce, 2000). The discrete approach starts with a numerical model of the primal equation and differentiates it directly line by line, or differentiates the numerical algorithm used to solve the continuous primal equation. One significant advantage of the discrete approach is that

the model building process can be automated, at least partially (Giering and Kaminski, 1998; Griewank, 2003). A variety of Automatic Differentiation (AD; also referred to as Algorithmic Differentiation) tools for various programming languages are available (e.g., http://www.autodiff.org/). It should be noted that the sensitivities from the discrete adjoint model are exact in the sense that they are the exact (to machine precision) first-order derivatives of the forward model unless approximations are made (Errico, 1997). It is expected that the sensitivities are comparable with those from the FDM (with properly chosen

perturbation sizes) or the CVM as the FDM and the CVM are both based on the same forward model (see Section 3 for the details of the CVM).

The continuous approach takes a governing equation, derives its adjoint equation, and numerically solves the adjoint equation. The continuous adjoint model is not constrained to using the same numerical scheme as the forward or the discrete adjoint

model (Sirkes and Tziperman, 1997). Take horizontal advection as an example. The forward equation and the adjoint equation share the same form; the only difference is that the adjoint equation runs backward in time (Hakami et al., 2007; Gou and Sandu, 2011). Implemented for advection in CMAQ is the Piecewise Parabolic Method (PPM), which is a higher-order Godunov-type method and uses intrinsic dissipation to improve stability and accuracy (Colella and Woodward, 1984; Byun and Schere, 2006). With the PPM employed for the adjoint equation, the corresponding continuous adjoint model bears the

same desirable numerical features.

## 2.2 The backward nature of an adjoint

Adjoint models are integrated backward in time. The nature of the backward propagation of adjoint sensitivities could be best demonstrated from a discrete perspective, which is detailed in Wang et al. (2001). Suppose we have the forward primal model written as

$$\boldsymbol{c_t} = \boldsymbol{G_{t-1}(c_{t-1})}, \tag{3}$$





where $\boldsymbol{c}$ is the vector of state variables (e.g., concentrations), subscript '$t$' indicates time, and $\boldsymbol{G}$ denotes the primal model.

Linearizing the above equation one can obtain the corresponding TLM (Talagrand and Courtier, 1987) as

$$\delta\boldsymbol{c_t} = G'_{t-1}\delta\boldsymbol{c_{t-1}}, \tag{4}$$

where $\delta\boldsymbol{c}$ represents perturbations to the state variables, and $G'_{t-1}$ is the Jacobian matrix. The Jacobian matrix is the first derivative of $\boldsymbol{G_{t-1}}$ with respect to the input vector and has the following form (Wang et al., 2001),

$$G'_{t-1} = \begin{pmatrix} \frac{\partial G_{11}}{\partial c_1} & \frac{\partial G_{12}}{\partial c_2} & \cdots & \frac{\partial G_{1n}}{\partial c_n} \\ \frac{\partial G_{21}}{\partial c_1} & \frac{\partial G_{22}}{\partial c_2} & \cdots & \frac{\partial G_{2n}}{\partial c_n} \\ \vdots & \vdots & \ddots & \vdots \\ \frac{\partial G_{n1}}{\partial c_1} & \frac{\partial G_{n1}}{\partial c_2} & \cdots & \frac{\partial G_{nn}}{\partial c_n} \end{pmatrix}_{t-1} . \tag{5}$$

The Jacobian matrix is not readily explicit in the model; by perturbing any one of the elements of the input vector $\boldsymbol{c}_0$, we obtain the values of the corresponding column of the Jacobian matrix. (Depending on the problem at hand, a combination of

perturbations could be useful and feasible.) The obtained sensitivities are with respect to the perturbed input of all output variables or a metric defined over the output variables.

To resolve the values of a row of the Jacobian matrix, the Jacobian matrix can be transposed to construct the following adjoint model,

$$\boldsymbol{\lambda_{t-1}} = \boldsymbol{G'^{T}_{t-1}\lambda_t}, \tag{6}$$

where $\boldsymbol{\lambda}$ is the vector of adjoint variables. The row of the Jacobian matrix holds the sensitivities with respect to all input variables.

For simulations from time 0 to t, the adjoint could be written

$$\boldsymbol{\lambda_0} = \boldsymbol{G'^{T}\lambda_t} = (\boldsymbol{G'_{t-1} \circ G'_{t-2} \circ \cdots \circ G'_0})^T\boldsymbol{\lambda_t} = \boldsymbol{G'^{T}_0 \circ G'^{T}_1 \circ \cdots \circ G'^{T}_{t-1}\lambda_t}, \tag{7}$$

where subscript '$t$' indicates the last time step used in calculating the Jacobian matrix. Because of the transposition, the order of the composition of the Jacobian matrices for different time steps is reversed. The Jacobian matrix at the last time step is applied first, instead of the one at the first time step. In other words, the adjoint sensitivities are propagated backward in time. A description from the continuous adjoint perspective of the backword propagation nature is found in Giles and Pierce (2000),

Hakami et al., (2007) or Henze et al. (2007).

Propagating backward in time introduces another prominent challenge for the adjoint model, i.e., the checkpointing of intermediate values of state variables for nonlinear processes. Unless the forward primal model is linear, the intermediate





values of the state variables are needed to calculate the transposed Jacobian matrix at each time step. Strategies of checkpointing are discussed in Section 2.4.

## 2.3 Automatic differentiation for the CMAQ science processes

The adjoint model development has been assisted with several AD tools. As mentioned earlier, the science processes in CMAQ include transport (advection and diffusion), gas-phase chemistry, aerosols and clouds. For the transport processes, TAMC (Tangent linear and Adjoint Model Compiler) was employed for the adjoint (Giering, 1999). The Kinetic PreProcessor (KPP) was adopted for the gas-phase chemistry and the aqueous chemistry of clouds (Damian et al., 2002). For aerosols and cloud dynamics, Tapenade was used to generate the adjoint (Hascoët and Pascual, 2013).

The AD approach is used for all the CMAQ science processes. The CMAQ code is written in Fortran 77/90 and is not ready for AD in general; significant modifications are required to process the original CMAQ code. For Tapenade 3.10, those modifications include preprocessing directives and macros, defining a proper cost function, constructing a root subroutine, and dealing with pointers and black-box subroutines, to name a few. Without the revisions, the AD tool either fails to generate the

adjoint files or produces one that requires excessive manual intervention.

In addition to the changes mentioned above, an important numerical procedure, the bisection method, needs some special treatment for the adjoint. The bisection procedure does not provide a passage for the propagation of sensitivity information. In the current work, the post-differentiation technique (Bartholomew-Biggs, 1998) is implemented before the adjoint is derived

so as to obtain sensitivity information in a manner consistent with the underlying algorithm. The extensively used bisection procedure remains in the CMAQ code; after the solution is converged, an extra step of the Newton-Raphson method is attached to the bisection procedure to facilitate calculation of the gradient of the equation at the root (Capps et al., 2012).

The adjoint models generated by the AD tools do not run right out of the box. A post-processing step is necessary to prepare

the code for testing, including modifications based on the error/warning messages issued by the tools and checking against the original CMAQ code, for example, for missing code parts. In general, the full cycle of development involves code preparation, AD, post-processing, and validation, and is repeated as many times as needed.

### 2.3.1 Aerosols

The CMAQ model uses the modal approach to treat aerosols (Binkowski and Roselle, 2003). Specifically, the size range of an

aerosol species is divided into three modes: the Aitken mode of aerosols with a geometric diameter less than 0.1 µm, the accumulation mode with a geometric diameter between 0.1 µm and 2.5 µm, and the coarse mode with a geometric diameter greater than 2.5 µm. Fine particles include those with a geometric diameter less than 2.5 µm, which include all particles in the





Aitken and accumulation modes. Although the geometric diameter is important for distinguishing the modes, it is a derived quantity in CMAQ. The aerosols are represented by mass, number and surface area by design; all the other quantities required for simulation are derived from these three representative moments based on the lognormal assumption of the size distributions (Binkowski and Roselle, 2003).

The key components of the CMAQ aerosol module include the production of secondary organic aerosols (SOA), new particle formation from nucleation, particle coagulation and condensational growth, heterogeneous chemical reactions to generate nitric acid, mode merging and aerosol thermodynamics (see Subsection 2.3.2 for cloud-related aerosol processes). Aerosol thermodynamics is treated with the ISORROPIA thermodynamic equilibrium model (Nenes et al., 1998), for which the adjoint,

ANISORROPIA, has been developed and documented in Capps et al. (2012). Bisection procedures are extensively used in ISORROPIA and the post-differentiation technique mentioned in Session 2.3 was employed to ensure the propagation of sensitivity information. Capps et al. (2012) discuss how the non-linearity and solution discontinuity in ISORROPIA poses great challenges in validating the adjoint model.

To generate the adjoint by AD, the rest of the processes in the aerosol module are lumped into a box model. The corresponding Fortran code is preprocessed for the AD tool. Although a Newton-type iterative procedure, instead of a bisection method, is used in the SOA process, the post-differentiation technique is implemented to improve computational efficiency. In other words, the original Newton routines are used to obtain a converged solution, which is then used to propagate the sensitivity information through the corresponding adjoint routine generated from the 1-step Newton method. It is worth noting that post-

differentiation is applied when post-processing the adjoint code; for AD, the original Newton routine is revised to iterate only once.

### 2.3.2 Clouds

The cloud module in CMAQ deals with cloud dynamics and aqueous chemistry. Depending on the cloud size, one of two solution techniques is employed. The resolved cloud sub-module (RESCLD) is invoked when the cloud size is larger than the

grid size. Under this circumstance, cloud dynamics become part of the transport process and need not to be treated separately in the cloud module. When clouds partially exist in a cell, the subgrid module for convective clouds (CONVCLD) is invoked, which resolves the vertical convective mixing in the boundary layer based on the Asymmetrical Convective Model (Pleim and Chang, 1992). The mixing process computes mixing ratios for each individual species inside and outside a cloud. The obtained mixing ratios are then redistributed to each layer according to its initial value using a weighting function. The nonlinearity

introduced by the weighting function has proven be problematic in differentiation as discussed in Section 3 during testing. Both sub-modules, RESCLD and CONVCLD, simulate in-cloud scavenging, wet deposition, and aqueous chemical reactions. An exponential decay formulation is used for the in-cloud scavenging processes. For cloud dynamics, the same AD tool and procedure as for the aerosol module are employed for the adjoint development.



For aqueous chemistry, KPP version 2.2.3 is used (Damian et al., 2002). Unlike the other AD tools used in this study, KPP operates on the algorithmic level instead of directly on the code; given a chemical mechanism, the KPP kinetic preprocessor can generate the corresponding computer code in several languages including Fortran 77/90. The KPP has the capability to

generate the forward model, the DDM/TLM, and the adjoint through separate runs. A detailed treatment of cloud chemistry with KPP can be found elsewhere (Fahey et al., 2017). The species treated in aqueous chemistry are consistent with the CB05 chemical mechanism and the AERO5 aerosol module in the current adjoint implementation.

### 2.3.3 Gas-Phase Chemistry

As done for the previous version of CMAQ-ADJ (Hakami et al., 2007), the KPP kinetic preprocessor is used to generate the

subroutines required for constructing the adjoint. The chemical kinetic mechanism implemented is updated to CB05 from CB-IV for the previous adjoint (Yarwood et al., 2005).

### 2.3.4 Transport

The transport module in CMAQ 5.0 consists of four components including horizontal advection, vertical advection, horizontal diffusion, and vertical diffusion. Horizontal advection is further divided into two $x$- and $y$-direction components with the order of the two advections alternating to maintain a symmetric form (Byun and Schere, 2006). As discussed in Subsection 2.1, the

PPM scheme is implemented for the advection process. PPM is monotonic and enforces positivity (Byun and Schere, 2006). As the sensitivity values could be either positive or negative, the positivity enforcing feature should be disabled when the method is applied for the adjoint equation to develop a continuous version of the adjoint (Gou and Sandu, 2011). The discrete adjoints of all the four components of transport were developed using TAMC version 5.3.2, which is an AD tool for Fortran-

77 programs with partial Fortran 90 support (Giering, 1999).

### 2.4 Manual interventions

Manual interventions are necessary to revise and assemble the adjoint source code generated by AD tools. First, the warning and error messages issued by AD tools are checked. Once the code is successfully compiled, the forward sweep of the adjoint routines is checked against the original CMAQ for completeness. For post-differentiation, the iterative Newton-Raphson

method originally implemented for SOA is added back to the forward sweep, to replace the one-step version created specifically for AD. The solution is saved (pushed into stack) after convergence and restored (popped out of stack) for the one-step adjoint routine of the Newton-Raphson method. Last, but not least, the adjoint code is refined according to the coding guidelines of CMAQ including indentation styles and the usage of macros and directives.

As mentioned in Section 2.2, the adjoint model is integrated backward in time and checkpointing is required for active variables used in the propagation of sensitivity information through nonlinear processes. There are two strategies to retrieve intermediate





values: repeatedly run the forward model or run the forward model once and store all the intermediate values. In practice, a trade-off between the two strategies is employed, as repeatedly running the forward model is prohibitively expensive and saving all intermediate values is impractical due to limited storage space (Wang et al., 2009). Intermediate values can be saved to memory, allowing for faster access, but subject to physical memory constraints.

The AD tools automatically employ a combined strategy of rerunning and checkpointing for intermediate values of active variables. For example, Tapenade performs a live analysis to determine if a variable is active and automatically apply a strategy that combines checkpointing and rerunning for those active variables at the subroutine level. Every adjoint routine contains a forward sweep and a backward sweep. During the forward sweep (rerunning), the values of active variables (including control

variables) are pushed into a stack (checkpointing), and then are popped out and used during the backward sweep. There are different stacks for different data types such as integer (for conditional/branch control), real, and double precision variables. The values of active variables are checkpointed to a stack of its own type accordingly. Stacks operate on the last-in first-out principle and are well-suited for the checkpointing purpose. Attention should be paid to the live analysis process; the "save" attribute or a bug in the AD tool could unexpectedly cause corrupted checkpointing of an active variable.

This implementation of internal checkpointing entails saving of numerous intermediate variables and requires far more computer memory than available for a typical regional air quality application with all the science processes involved for a multiple-day simulation. The strategy adopted for the current full adjoint model is the same as the one used for the previous version (Hakami et al., 2007). A forward run is carried out with the original CMAQ model revised to checkpoint to files the

values of active variables for each nonlinear science process at every synchronization time step (i.e., the external time step for horizontal advection). The backward adjoint run then reads the checkpointing file at the beginning of the corresponding science process.

## 2.5 Adjoint forcing pre-processor

The adjoint model calculates the sensitivity of a cost function, $J$, with respect to model parameters. The adjoint forcing, $\varphi_i$, in

Equation 2 corresponds to $\frac{\partial J}{\partial c_i}$. Like emissions in the forward sweep, the adjoint forcing is applied at every time step in the backward sweep and may be applied in any grid cell. Typically, the cost function depends on the concentration-based results of the forward sweep. The derivative of the cost function with respect to the concentration of interest must be calculated for each time step and added to an IOAPI-compliant netCDF file. To ease the burden of introducing different cost functions for users, a Python-based adjoint forcing pre-processor has been developed. As a model, the pre-processor includes the calculation

of the local maximum daily 8-hour average ozone concentration, which is a regulatory metric in the U.S. This example addresses issues of shifting to local time and a forcing dependent on an average in every grid cell. The example produces an adjoint forcing file corresponding to this cost function ready for use in the backward sweep. The Python-based pre-processor



will make implementation of additional cost functions such as 24-hour average aerosol constituents or observation operators for satellite-based atmospheric composition straightforward.

## 3 Model Evaluation

The adjoint model is evaluated on a process-by-process basis against the brute force Finite Difference Method (FDM) and the Complex Variable Method (CVM) (Squire and Trapp, 1998). Box or column models are used when applicable to maximize the number of comparison pairs from the backward and forward sensitivity test runs (Hakami et al., 2007). The finite difference method is straightforward to implement and has been used for evaluating other sensitivity methods (Dunker et al., 2002; Napelenok et al., 2006; Hakami et al., 2007; Henze et al., 2007), but in some of the cases requires finding optimal step sizes to obtain accurate results. The searching process is time-consuming and could be impractical when a large data set such as that of a three-dimensional atmospheric model is involved. With the CVM, the results are practically insensitive to the perturbation size with the exception of some rare circumstances discussed in Section 3.2.1.

The air quality simulation scenario used for the evaluation is for the contiguous U.S. domain with a 36-km horizontal resolution and 24 vertical layers for the first seven days of April 2008. More details about meteorological inputs, initial and boundary conditions, and emissions are provided in Turner et al. (2015a).

### 3.1 The complex variable method

The CVM for the first-order sensitivities is formulated as follows, which is a Taylor series expansion about an imaginary perturbation step,

$$J(p + ih) = J(p) + J'(p)(ih) - J''(p)h^2/2 + \cdots \tag{8}$$

where, J is the cost function, $p$ is the parameter to which the gradient is evaluated, $i$ the imaginary unit, and $h$ is the perturbation step.

Extracting only the imaginary part and rearranging,

$$J'(p) = \Im(J(p + ih))/h + O(h^2) \tag{9}$$

where, $\Im$ represents the operator to extract the imaginary part of a complex number. As seen in Equation 8, the CVM has a second-order accuracy, which is the same as the central finite difference formulation. The CVM, however, is not subject to subtractive errors and therefore permits the use of as small a step size as allowed in floating-point calculations to achieve much better accuracy, which helps in situations when the brute force FDM fails or proves inaccurate.





To construct a CVM version of CMAQ, several guidelines are followed which include changing the data types of all the active variables from REAL/DOUBLE PRECISION to the corresponding complex type, creating a complex version of intrinsic functions such as MAX, MIN and ABS, and evaluating only the real part of complex variables used in conditionals (Giles and Piece, 2000). The original 3D CMAQ framework is set up for testing with the processes or sub-processes under investigation

enabled and the rest commented out. To run the CVM, a perturbation is added to the imaginary part of a source variable at a time step of interest and then the imaginary part of a receptor is extracted and divided by the perturbation size to obtain the CVM sensitivity.

### 3.2 Process-by-process model evaluation

#### 3.2.1 Aerosols

As mentioned before, the CMAQ aerosol module incorporates the following science processes: SOA formation, nucleation, condensation, coagulation, heterogeneous chemistry, mode merging, and aerosol thermodynamics. The sub-processes are evaluated individually and eventually as a whole in simulations in which other processes (e.g., advection) are turned off. Adjoint sensitivities are first compared with those from the FDM, and if a mismatch persists, the CVM method is implemented (if feasible) for that process for further evaluation.

Figure 1 shows the adjoint (ADJ) and CVM sensitivities of an example SOA process. The sensitivities are of the final concentrations of an accumulation-mode aerosol species, AALKJ ($\mu g/m^3$), with respect to the initial concentrations of a semi-volatile species, SV_ALK (ppmV), from a one-day test run. For this process, the FDM behaved well for most of the test cases (results not shown); in the few cases when the ADJ and FDM did not agree and tuning with the perturbation sizes did not help,

use of CVM demonstrated good accuracy of adjoint results (i.e., agreement along one-on-one line). This is an example of numerous cases where FDM was found to be inaccurate or inadequate in evaluating adjoint sensitivities. It should be noted that the adopted perturbation size for the CVM is generally $10^{-12}$; a smaller perturbation size usually does not improve accuracy of the obtained sensitivities, but risks diminishing the sensitivity information through propagation due to the single precision nature of some of the variables within CMAQ. A smaller perturbation size could be used if double-precision data types were

adopted for the whole code. For the SOA process, however, a smaller perturbation size of $10^{-24}$ does improve the accuracy. Test results of the other organic aerosol species show similar accuracy.

Generally good spatial agreement is observed between the ADJ and the CVM when all aerosol sub-processes except thermodynamics (ISORROPIA) are included in a one-day run (Figure 2). The cost function is the final concentrations of an

accumulation-mode aerosol species ASO4J ($\mu g/m^3$) and the perturbation variable is the initial concentrations of an Aitken-mode aerosol species ASO4I ($\mu g/m^3$) with a perturbation of $10^{-12}$ for the CVM. One of the reasons to choose the two model species for testing was that sulphate aerosol is a crucial component of fine aerosols. The other reason was that the pathway



from ASO4I to ASO4J covers the size range for the important processes of coagulation and the numerical mode merging that handle interactions between the Aitken and accumulation modes.

With the addition of the aerosol thermodynamics (ISORROPIA/ANISORROPIA), the degree of agreement is degraded (Figure 3, bottom left panel). The disagreement is likely caused by the discontinuities abundant in the solution surfaces of ISORROPIA, which at the code level is manifested as a series of executive branches (Capps et al., 2012). As shown in Equation 8, the introduction of the imaginary part, $ih$, changes the value of the real part of a complex variable. Although the change is of $O(h^2)$, different executive branches can be invoked as a result. When the ADJ and CVM sensitivities from the same branches are compared, the agreement is much improved (Figure 3, right panel).

As an example for the full aerosol model application, the sensitivities of final ASO4J ($\mu g/m^3$) with respect to initial NH3 (ppmV) from a one-day simulation are evaluated against CVM estimates (Figure 4). Although the branches in ISORROPIA are made identical between the ADJ and the CVM, the agreement is worse than what is seen in Figure 3; reducing the perturbation size does not affect the agreement. One possible explanation is the high nonlinearity inherent within the aerosol processes including ISORROPIA that renders the CVM with a finite perturbation size ineffective in producing accurate sensitivities. A detailed discussion of the discontinuities and nonlinearity of ISORROPIA is given in Capps et al. (2012). Overall, the sensitivities generated from the aerosol adjoint show acceptable accuracy when compared with the CVM.

### 3.2.2 Cloud dynamics and chemistry

The adjoint of mixing due to formation of sub-grid convective clouds was tested by comparing FDM sensitivities (CVM was not available for all cloud processes) to adjoint-calculated sensitivities in a one-day test run. The test, shown in Figure 5 for a perturbation of 0.01 $\mu g/m^3$ in initial ASO4J, was successful but exhibited sensitivity to the type and size of the perturbation as discussed below. For the FDM, either a percentage perturbation or a perturbation with an absolute small value can be used. For test cases with small initial concentration values, however, percentage perturbations are sometimes not detectable at the end of run (due to round-off errors), which leads to diminished sensitivity values. The non-linearity introduced by the weighting functions implemented for re-distribution of gas and aerosol concentrations in convective mixing made this issue stand out; although not apparent in Figure 5, the FDM constantly failed to produce some larger values observed in the adjoint sensitivity field. On the other hand, perturbations with an absolute value tend to undermine the accuracy of the FDM sensitivities by causing truncation errors that are comparable to the size of perturbation. Absolute perturbations have been generally favoured and used in the evaluations of cloud processes. The adjoint of the resolved clouds module shows a better agreement with the FDM (not shown) than that of the convective clouds.

The adjoint of the aqueous chemistry science process was tested similarly, with a 0.01 $\mu g/m^3$ perturbation in initial ASO4J. Figure 6 shows a successful test, with close agreement between the ADJ and the FDM sensitivities. Aqueous chemistry was





further tested by examining the sensitivity of final ASO4J to small (0.01/0.001/0.0001 ppb) perturbations in gas-phase $SO_2$ and results are shown in Figure 7. Mismatches were apparent in all three plots between the ADJ and the FDM, especially the column with high values of FDM sensitivities and zero ADJ sensitivities. As discussed above, small initial values could render the FDM approximation difficult. A way to quickly check whether these small values are causing disagreement is to semi-

normalize the sensitivities by multiplying with the initial values. The results from semi-normalization with initial $SO_2$ concentrations confirm that the deviations were caused by small initial values (Figure 7, bottom).

### 3.2.3 Gas-phase chemistry

ADJ and FDM sensitivities of the final $O_3$ concentrations (ppmV) with respect to the initial $NO_2$ concentrations (ppmV) from a one-day test with the updated CB05 chemical mechanism show good agreement, depending on the choice of appropriate

perturbation size (Figure 8). Results of three perturbation sizes 0.1, 0.01 and 0.001 ppb are shown for the FDM, which demonstrate the impact of perturbation sizes on accuracy.

Mismatches in some test cases led to the development of a CVM for the gas-phase chemistry. During the testing, Jacobians of ADJ and CVM sensitivities are created to provide a means for visual examination of all gas-phase species. An example is

shown in Figure 9, where the *x* and *y* axes represent all species involved in gas-phase chemical reactions, and each point represents the sensitivity of an x-axis species with respect to a y-axis species. Presented in Figure 10 is the corresponding scatter plot which compares the ADJ with the CVM and shows an excellent match between the two methods. The absolute sensitivity values are used in the tile plot in Figure 9 for better visualization.

### 3.2.4 Transport

The process of transport includes advection and diffusion. For the advection process, the nonlinear PPM scheme is implemented in CMAQ as discussed in 2.3.4.

The adjoint of the advection equation could be written as,

$$-\frac{\partial \lambda_i}{\partial t} = \nabla \cdot (\mathbf{u}\lambda_i). \qquad (10)$$

Compared to the corresponding terms in Equation 1, the signs of the two terms in the above equation are reversed, which implies that the adjoint values are propagated in an opposite direction (Giles and Piece, 2000). With a reversal in wind direction, the PPM could be used to integrate the adjoint equation backward in time and solve for continuous adjoint sensitivities.

The discrete adjoint of advection is the result of direct differentiation of the numerical model and can be validated against the

FDM/CVM component by component. Figure 11 of horizontal advection in *x* direction shows a good agreement between the discrete adjoint and the CVM of sensitivities of final ASO4J (µg/ m³) with respect to initial ASO4J (µg/m³). However,



numerical noises are clearly in sight; unfortunately, such spurious oscillations from discrete adjoints derived from a nonoscillatory advection scheme are not uncommon and a desirable fix does not appear possible (Thuburn and Haine, 2001).

For the continuous adjoint (CADJ), the horizontal and vertical advection processes were tested as a whole, as in forward
CMAQ these processes are linked together for mass conservation and consistency between transport processes in CMAQ and the underlying meteorological model. One issue with testing the advection processes altogether is that pointwise comparison of sensitivities becomes much more computationally expensive as the models are not row or column models anymore, and running the adjoint and the FDM/CVM once would generate only one pair of sensitivities for comparison. To partially remediate this situation, we defined the cost function for the adjoint as the final average ASO4J concentration across the entire
surface layer (instead of concentration at a single cell which would lead to a small number of cells with sensitivity signals over time, not sufficient for validation), and then randomly selected a number of cells at the surface for the CVM runs. The CADJ and the CVM agree well as shown in Figure 12 where the regression line has a slope close to unity and $y$-intercept close to zero, and the value of $R^2$ is 0.957.

The choice between the continuous and discrete adjoint would depend on the type of problem at hand. For instance, the
continuous adjoint is generally desirable when performing backward sensitivity analysis as an oscillating sensitivity field (visible in Figure 11) may defy physical justification (Hakami et al., 2007; Henze et al., 2007). For optimization problems, the discrete adjoint would be preferable as it produces exact gradients (subject to round-off errors) that could help the optimization process converge (Giles and Pierce, 2000). However, it was reported that the noisy gradient field obtained from the discrete
adjoint could cause the optimization to converge to local minimums (Vukićević et al., 2001). The continuous adjoint may outperform the discrete in terms of computational efficiency and accuracy, as found by Gou and Sandu (2011) with their 4D-Var experiments.

Air pollutant emissions are processed in vertical diffusion in CMAQ. As shown in Figure 13a, the adjoint of vertical diffusion
compares well with the finite difference method (Figure 13a). The adjoint of emissions also works well as demonstrated in Figure 13b, where the adjoint sensitivities of final $NO_2$ concentrations (ppmV) to initial $NO_2$ emissions (moles/s) compare favourably with the FDM.

**3.3 Full-model evaluation**

For the full adjoint model, interactions between cells through transport make it prohibitively costly to generate sufficient
sensitivity pairs for an extensive comparison as conducted for the process-by-process evaluation with box or column models. Presented in Table 1 are the FDM, CVM, adjoint with continuous treatment for transport and adjoint with discrete treatment for transport (DADJ) sensitivities obtained for a few grid cells.





In general, the adjoint agrees well with the CVM, especially the DADJ with a relative error less than 10%. The FDM sensitivities with a 10% perturbation step on the other hand are not quite in accordance, which is why a full CVM was created. The problem with the FDM has been discussed earlier and is not repeated here.

Results shown in Table 1 suggest that the discrete adjoint has a better agreement with CVM than the continuous adjoint. However, it is important to note that better agreement between the discrete adjoint and CVM should not be understood as better accuracy of the discrete adjoint in comparison with continuous adjoint. The numerical solution to the advection equation entails inherent truncation errors from discretization schemes. These errors exist in solving the forward or adjoint advection equations; however the discrete adjoint by design remains loyal to, and consistent with the errors in the forward application

(CVM in this case), while numerical solution to the continuous adjoint will result in different and inconsistent errors. The continuous adjoint is a different representation of the impacts on the adjoint cost function, but of similar mathematical accuracy, when compared to the forward or tangent linear model; therefore, the numerical solution to it should be considered as accurate as the discrete adjoint, regardless of the agreement with forward-based benchmarks such as CVM.

### 3.4 Computational system requirement

Adjoint simulations entail a significantly higher computational demand than forward CMAQ. First, the checkpointing files required for the adjoint simulations need a significant amount of storage space. For each science process the amount of storage can be estimated as $N_c \times N_r \times N_l \times N_s \times N_t \times N_b$ bytes, where $N_c$, $N_r$, $N_l$, $N_s$, $N_t$, and $N_b$ represent the numbers of columns, rows, layers, chemical species, synchronization time steps, and bytes for a single-precision number ($N_b = 4$), respectively. For our computational domain with 148 columns, 112 rows, 24 vertical layers and 12-minute synchronization (i.e., 120 time

steps/day), the checkpointing file for each adjoint simulation for aerosols with 137 chemical species takes approximately 24 GB storage space for a day. For the other science processes the sizes of checkpointing files are approximately as follows: clouds, 24 GB; chemistry with 96 species, 17 GB; vertical diffusion with 1 layer, 1 GB; the continuous adjoint of horizontal and vertical advections with 1 species, 0.2 GB for each process. For a one-month adjoint simulation, the checkpointing files occupy about 2 TB of storage space, which is about 10 times the storage needed for typical CMAQ simulations. For higher

resolution simulations, the storage needs would increase proportionally due to the number of grid cells, but also due to increasingly smaller time steps dictated by the Courant number for smaller horizontal grid sizes. For very high resolution simulations (e.g., 1 km horizontal resolution), the required checkpoint storage space can be as large as 1 TB/day. To mitigate the burden on storage, it is plausible to run the adjoint segment by segment, i.e., by generating the checkpointing files only for a few days at a time when running the forward CMAQ model. Since the adjoint runs backward in time, this strategy works at

the expense of CPU time as the forward model must be repeated.

Adjoint simulations also require significantly higher CPU times, as the re-computation of intermediate values of adjoint simulations as discussed in Section 2.4 is an additional computational overhead. Furthermore, the significant amount of





input/output (IO) operation associated with the checkpointing, leads to additional CPU time and can result in noticeable loss of computational efficiency in systems. Typically, the adjoint simulation takes approximately 4 times as long as the forward CMAQ. Intensive IO for checkpointing files can also result in reduced scalability of the adjoint model, as the IO libraries currently implemented are serial.

5   **4 Model Application: Backward Sensitivity Analysis**

To demonstrate a policy-relevant application of the multiphase adjoint of CMAQ, we estimate the impacts of emission sources on adverse $PM_{2.5}$ health impacts across the US in an adjoint sensitivity analysis; more details of an adjoint-based approach for source attribution of health impacts can be found elsewhere (Pappin et al., 2013 and 2016). Mortality counts, or nationwide valuation of mortality induced by air pollution is a scalar well suited for formulating as the adjoint cost function. We define 10  an adjoint cost function, J, which represents the monetized valuation of annual deaths due to long-term $PM_{2.5}$ exposure within the U.S., as below,

$$J = \sum_{x,y} V_{SL}\, M_{0,x,y}\, P_{x,y} \left( 1 - e^{-\beta\, C_{x,y}} \right) \qquad\qquad (11)$$

where J (\$/yr) is calculated using the location-specific baseline mortality rate ($M_{0,x,y}$; $yr^{-1}$) and population ($P_{x,y}$); $\beta$ as the risk 15  estimate that represents the slope of the log-linear concentration-mortality curve; the time-averaged and location-specific $PM_{2.5}$ concentration, $C_{x,y}$ ($\mu g/m^3$); and the value of a statistical life, $V_{SL}$ for monetizing outcomes. We use a $\beta$ of 0.5827% per $\mu g/m3$ from Krewski et al. (2009) for adults aged 30-99 and for deaths due to all causes. We use the simulation period-average $PM_{2.5}$ concentration in place of the annual average due to our shorter episode duration, while recognizing the limitations of extrapolating from April to the full year.

The use of this cost function results in gradient estimates that provide a measure of how reducing emissions by 1 metric ton would result in societal benefits across the U.S., referred to as marginal benefits or benefits-per-ton (BPT, \$/yr). Benefits-per-ton for primary $PM_{2.5}$ emitted across source locations in the US are shown in Figure 14, as well as for the $PM_{2.5}$ inorganic precursors $NH_3$, $SO_2$, and NO. NO sensitivities are negative in some regions with more frequent $NO_x$-inhibited regimes, mainly 25  due to the role that ozone plays in night time nitrate formation. BPTs show a great deal of spatial variability but generally follow the population distribution for primary $PM_{2.5}$ emissions, while for inorganic precursor emissions areas of higher influence are dictated by transport patterns, secondary (inorganic) aerosol formation dynamics, and lifetime of secondary particles. In other words, BPTs are generally highest in emission locations that have large potential for affecting downwind population centers.

BPTs values such as those shown in Figure 14 have the potential to form important quantitative decision metrics, as they provide a means to squarely compare societal benefits of emission reductions with control costs associated with those

reductions. It is worth emphasizing that given the short simulation period, these results should not viewed as BPT values that can be used in policy development and benefit assessment, instead they are meant to serve as a demonstration of the utility and efficacy of the adjoint model to attribute health impacts to individual sources.

## 5 Conclusions

In this paper, we develop a multiphase adjoint of CMAQ. A rigorous point-to-point evaluation against the brute force FDM and CVM is conducted for each individual process and the full model with all processes included. Overall, the adjoints appear to produce sensitivities comparable to those generated by either the FDM or the CVM. The choice of the discrete or continuous version of the advection adjoint would depend on the type of problem to be solved. The continuous adjoint is preferred if the sensitivity field itself is of interest, as spurious oscillations would create intricate obstacles for exploring the underlying physical significances. For gradient-based optimization and data assimilation, the discrete adjoint might be advantageous for faster convergence but could risk the minimization settling upon some local minima. Some components of CMAQ that do not yet have an adjoint include the calculations of dry deposition velocities in vertical diffusion and photolysis rates in gas chemistry. The development of an adjoint for these two components is not considered essential. The CMAQ adjoint provides backward sensitivity analysis capabilities for a widely used model with detailed aerosol treatment, and enables a range of applications data assimilation, emission inversions, policy analysis, and source attribution of health impacts.

## 6 Code Availability

Upon completion of expanded user testing, the CMAQ Adjoint code will be hosted and distributed by the U.S. EPA.

## 7 Author Contribution

SZ contributed to the development of aerosol dynamics, clouds adjoint, aqueous chemistry, coordinated adjoint code assembly and testing, and prepared the manuscript. AH led the team and developed chemistry module. SC developed the ANISORROPIA, and contributed to development of the forcing generator and preparation of user manual. MT contributed to the development of the adjoint of aerosol dynamics. PP developed the adjoint of transport. KF provided the KPP configuration of aqueous chemistry. JR developed the checkpointing modules. MR developed the adjoint forcing generator. SH contributed to the debugging of vertical transport. DH, AR, AN, TC, CS, and GC provided feedback on adjoint development. AP performed adjoint health impact analysis. SN and JB provided support for existing and new CMAQ modules. All co-authors contributed to the preparation of the manuscript.



# 8 Acknowledgements

Hakami acknowledges support from the Natural Sciences and Engineering Research Council of Canada Grant RGPIN-2016-06181, Health Canada Contract 4500383370, and the Health Effects Institute Agreement 4962-RFA17-2/18-4. Henze received support from NASA grant NNX16AQ26G. Simulations were performed on computational resources of Compute Canada and Compute Ontario.



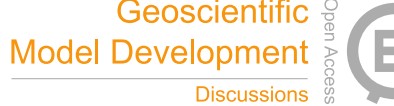

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



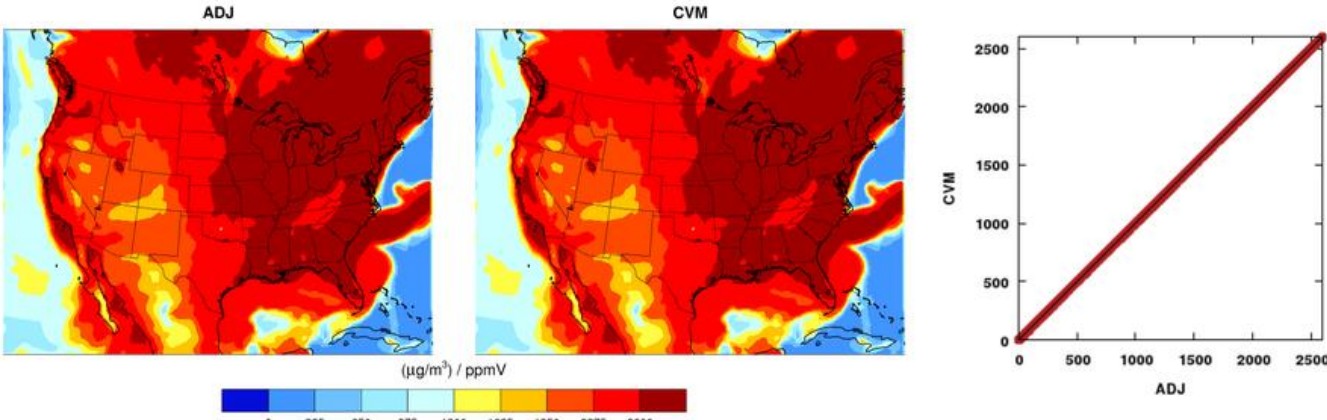

**Figure 1: Evaluation of the SOA (Secondary Organic Aerosol) process with the ADJ (Adjoint) against the CVM (Complex Variable Method) sensitivities of the final concentrations of an accumulation-mode aerosol species AALKJ ($\mu g/m^3$) with respect to the initial concentrations of a semi-volatile species SV_ALK (ppmV) from a one-day test run. The perturbation size for the CVM is 1.E-24.**

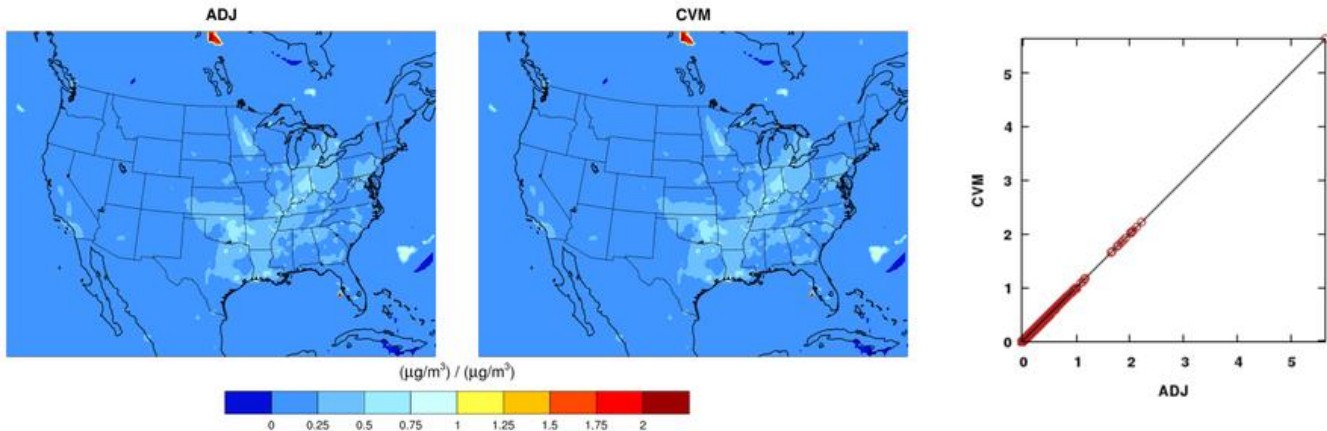

**Figure 2: Evaluation of all aerosol sub-processes but thermodynamics with the ADJ (Adjoint) against the CVM (Complex Variable Method) sensitivities of the final concentrations of an accumulation-mode aerosol species ASO4J ($\mu g/m^3$) with respect to the initial concentrations of another Aitken-mode aerosol species ASO4I ($\mu g/m^3$) from a one-day test run. The perturbation size for the CVM is 1.E-12.**



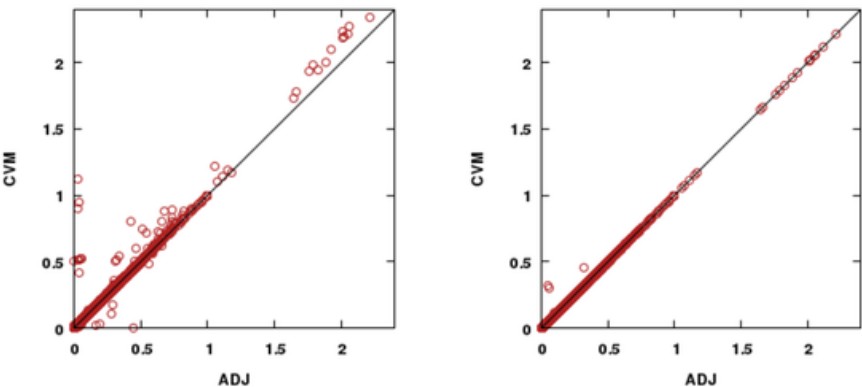

**Figure 3: Evaluation of all aerosol sub-processes with the ADJ (Adjoint) against the CVM (Complex Variable Method) sensitivities of the final concentrations of an accumulation-mode aerosol species ASO4J (µg/ m³) with respect to the initial concentrations of the Aitken-mode aerosol species ASO4I (µg/m³) from a one-day test run: left, original results; right, ISORROPIA branches set consistent between the CVM and the ADJ. The perturbation size for the CVM is 1.E-12.**

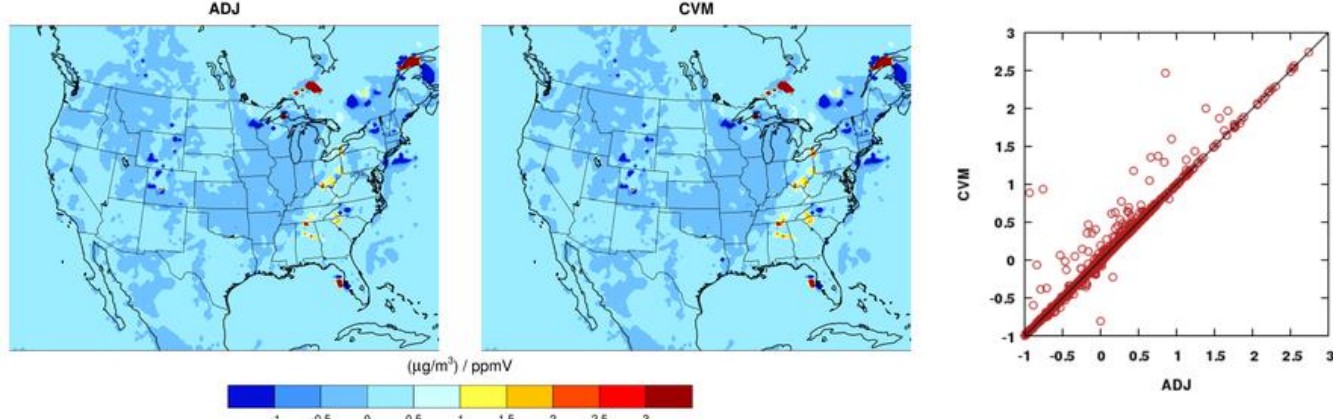

**Figure 4: Evaluation of all aerosol sub-processes with the ADJ (Adjoint) against the CVM (Complex Variable Method) sensitivities of the final concentrations of an accumulation-mode aerosol species ASO4J (µg/m³) with respect to the initial concentrations of NH3 (ppmV) from a one-day test run. The perturbation size is 1.E-12.**



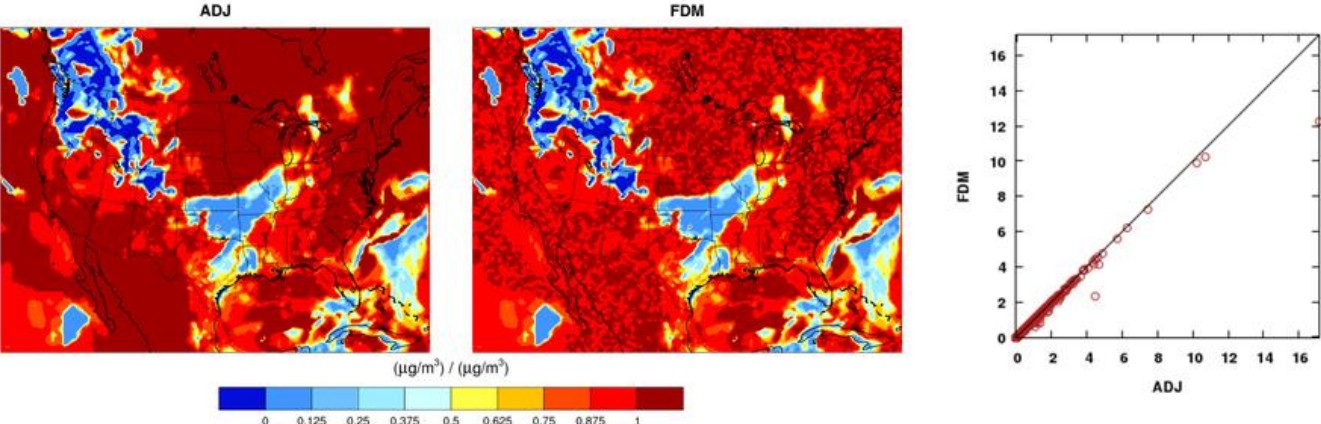

**Figure 5: Evaluation of sub-grid cloud mixing with the ADJ (Adjoint) against the FDM (Finite Difference Method) sensitivities of the final concentrations of an accumulation-mode aerosol species ASO4J (µg/m³) with respect to the initial ASO4J (µg/m³) from a one-day test run. The perturbation size for the FDM is 0.01µg/m³.**

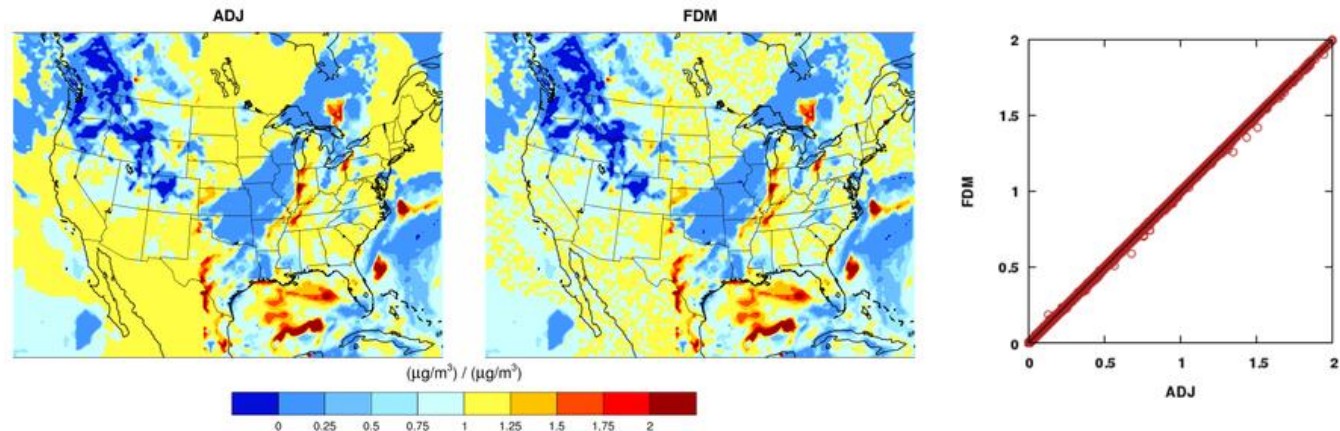

**Figure 6: Evaluation of cloud dynamics and aqueous chemistry with the ADJ (Adjoint) against the FDM (Finite Difference Method) sensitivities of the final concentrations of an accumulation-mode aerosol species ASO4J (µg/m³) with respect to the initial ASO4J (µg/m³) from a one-day test run. The perturbation size for the FDM is 0.01µg/m³.**







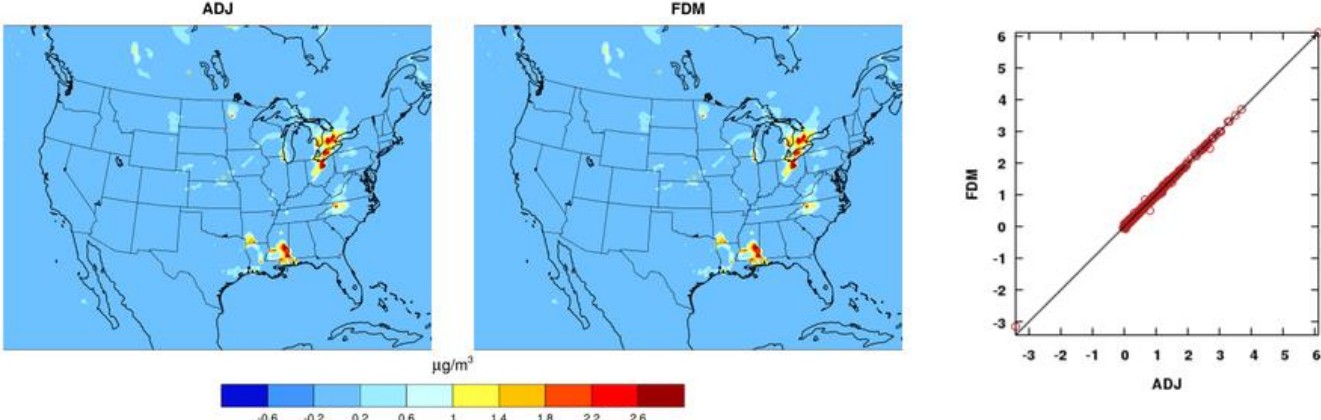

**Figure 7: Evaluation of cloud dynamics and aqueous chemistry with the ADJ (Adjoint) against the FDM (Finite Difference Method) sensitivities of the final concentrations of an accumulation-mode aerosol species ASO4J ($\mu g/m^3$) with respect to the initial SO2 (ppmV) from a one-day test run. The perturbation size for the FDM is 0.01/0.001/0.0001 ppb for the top three figures. The bottom figure is of sensitivities semi-normalized by the initial conditions of SO2 and the perturbation size for the FDM is 0.001 ppb.**





**Figure 8: Evaluation of gas-phase chemistry with the ADJ (Adjoint) against the FDM (Finite Difference Method) sensitivities of the final O₃ concentrations (ppmV) with respect to the initial NO₂ concentrations (ppmV) from a one-day test run. The perturbation sizes are 0.1, 0.01 and 0.001 ppb for the FDM in the plots from the top to the bottom.**



**Figure 9: The Jacobians of absolute sensitivities from the ADJ of gas-phase chemistry at a grid cell from a one-step run. The X and Y axes represent all species involved in gas-phase chemistry.**



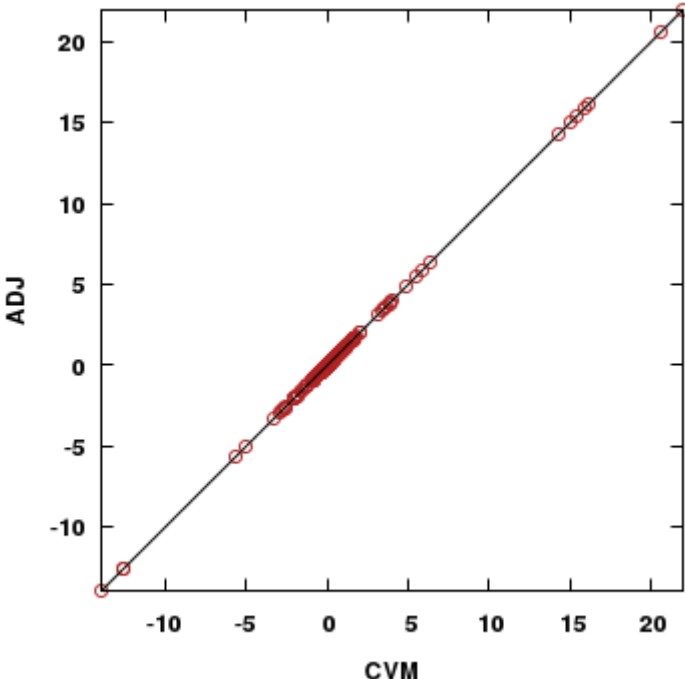

**Figure 10: Evaluation of gas-phase chemistry with the Jacobians of absolute sensitivities from the ADJ (Adjoint) and the CVM (Complex Variable Method) at a grid cell from a one-step run. A circle represents a pair of the absolute sensitivities from the ADJ (shown in Figure 9) and the CVM.**

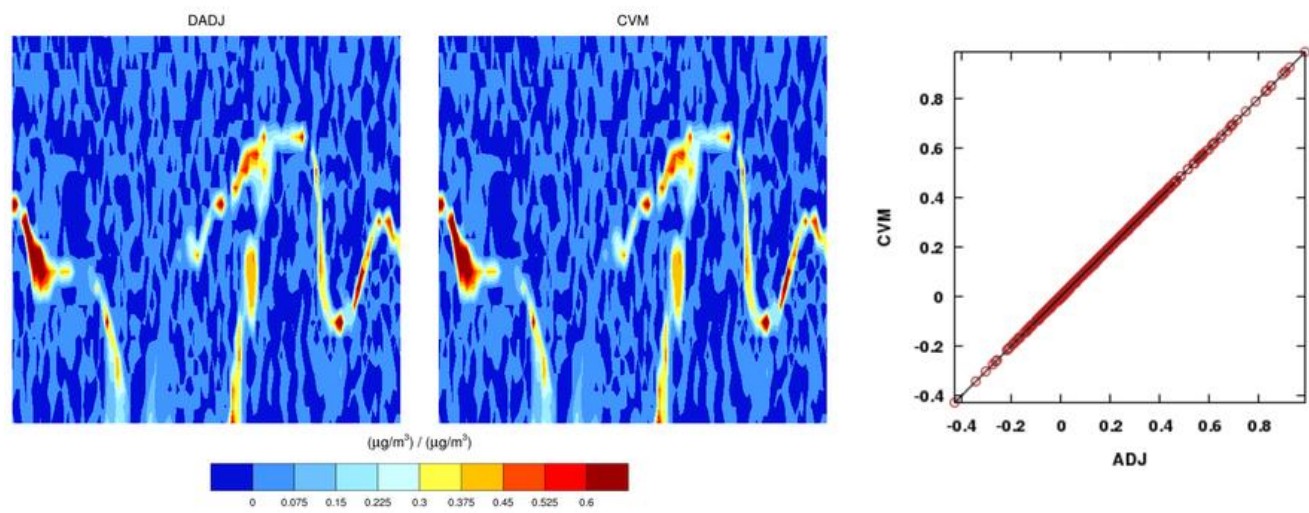

**Figure 11: Evaluation of horizontal advection in the *x* direction with the DADJ (discrete adjoint) against the CVM (Complex Variable Method) sensitivities of the final concentrations of an accumulation-mode aerosol species ASO4J (µg/m³) with respect to the initial ASO4J (µg/m³) from a one-day test run. For the top plots, the *x*- and *y*-axes represent the horizontal *y* direction and the vertical layers, respectively. The perturbation size for the CVM is 1.E-12.**



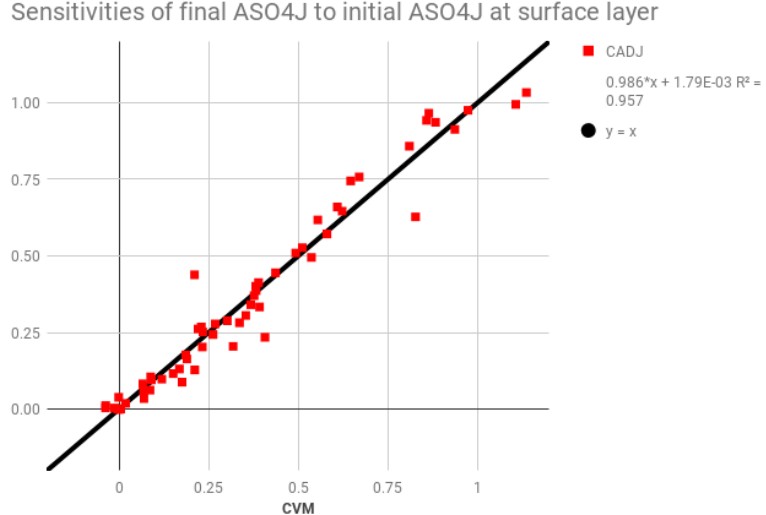

**Figure 12: Evaluation of advection with the CADJ (continuous adjoint) against the CVM (Complex Variable Method) sensitivities of the final concentrations of an accumulation-mode aerosol species ASO4J ($\mu g/m^3$) with respect to the initial ASO4J ($\mu g/m^3$) from a one-day test run. The perturbation size for the CVM is 1.E-12.**

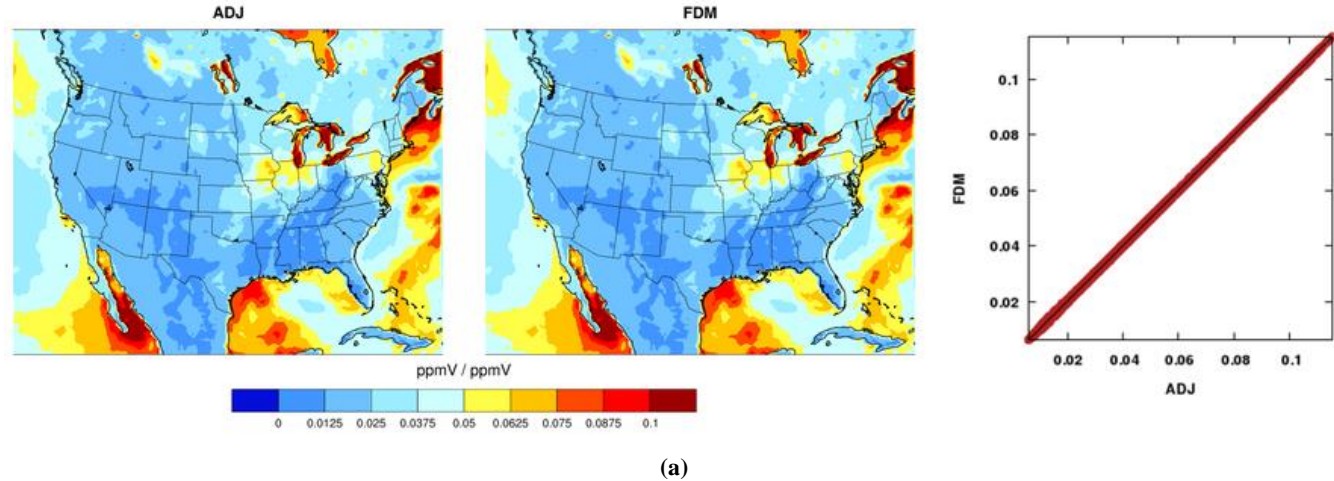

**(a)**



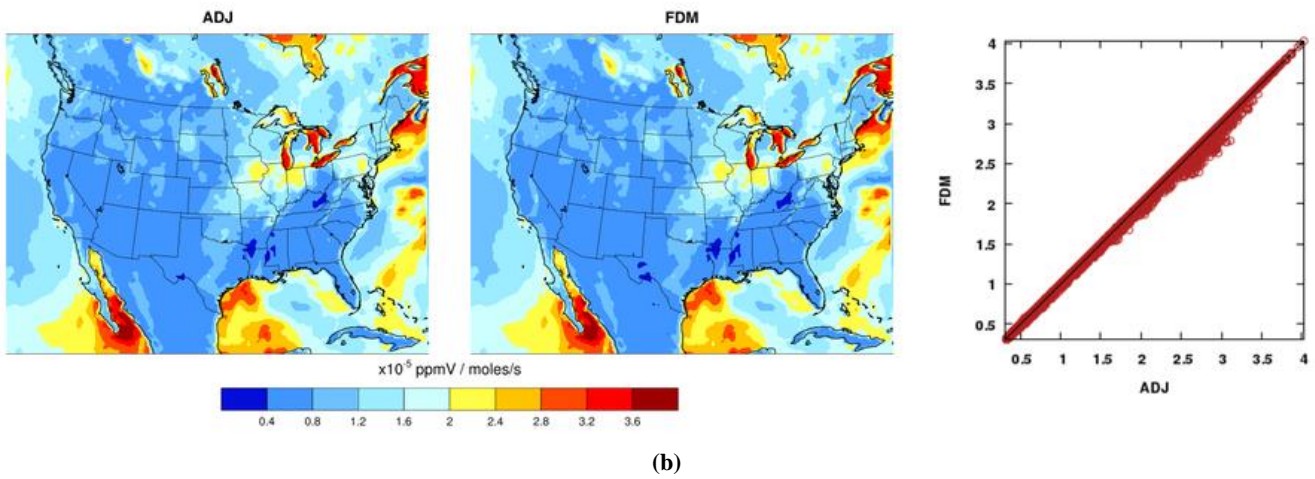

**(b)**

**Figure 13: Evaluation of vertical diffusion with the ADJ (Adjoint) against the FDM (Finite Difference Method) from a one-day test run: (a) sensitivities of the final ozone (ppmV) with respect to the initial ozone (ppmV) with a 1-ppb perturbation size for the FDM;**

5 **(b) sensitivities of the final nitrogen dioxide (ppmV) with respect to the initial nitrogen dioxide emissions (moles/s) with a perturbation size of 0.1 moles/s. The values in (b) are multiplied by $10^5$.**

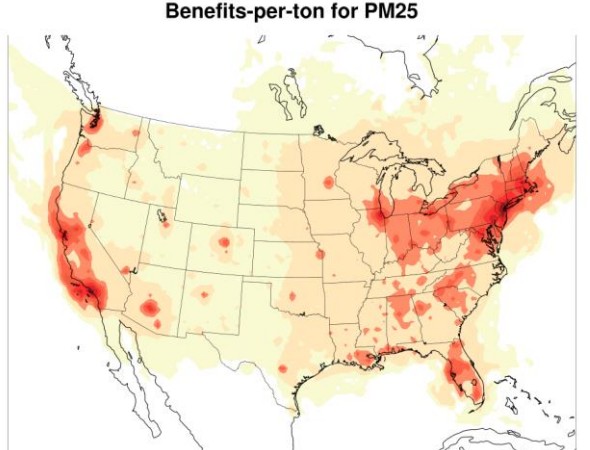
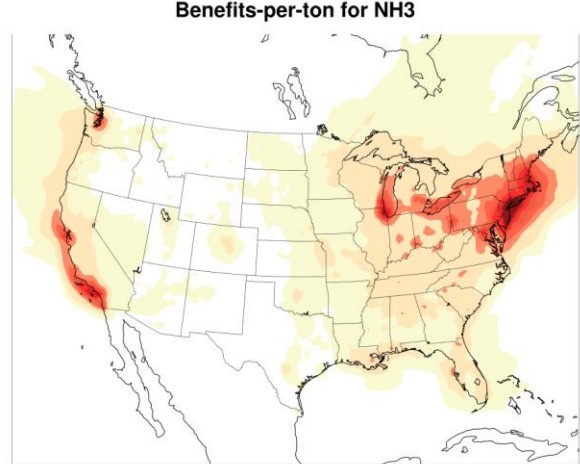



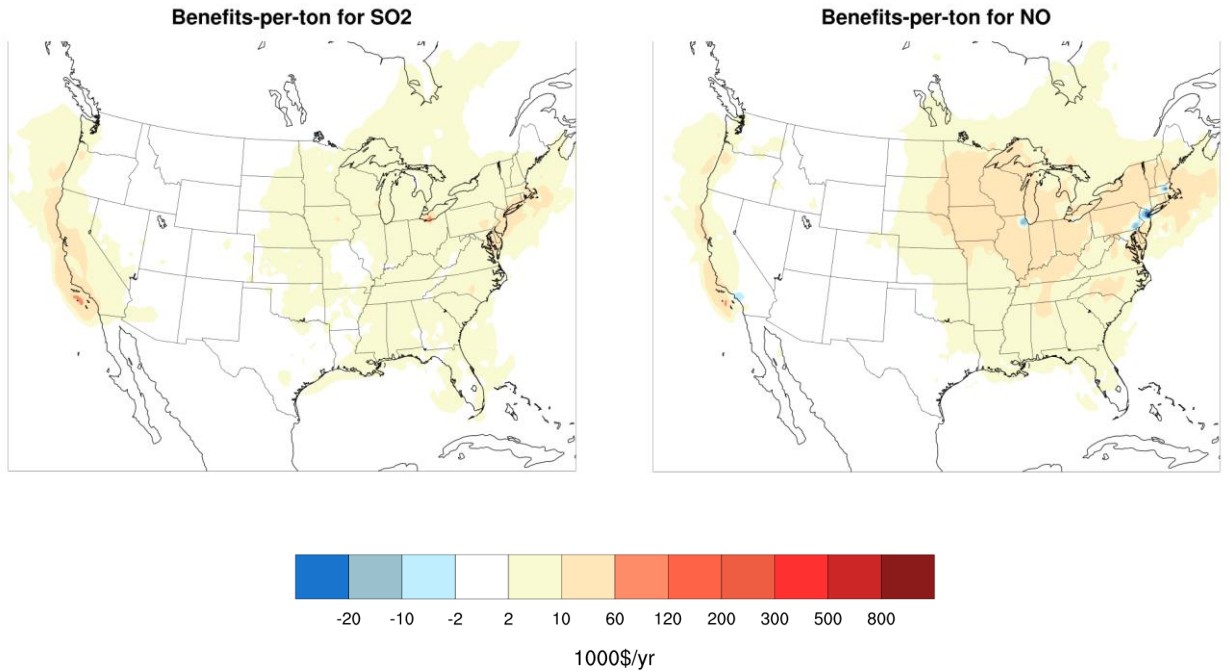

**Figure 14: Application of the adjoint model for adjoint sensitivity analysis to estimate the benefits-per-ton (BPT) related to long-term PM$_{2.5}$ exposure for primary PM$_{2.5}$ emissions in the US and its precursors NH$_3$, SO$_2$, and NO. The BPT is time-integrated and location-specific, i.e., each value in the figures represent the BPT for the specific emissions at the specific location. For example, a value of \$30,000/ton for SO$_2$ emissions at a location suggests that reducing emission of SO2 at that location entails \$30,000 in valuated benefits for the U.S.**

| Cells | #1 | #2 | #3 | #4 | #5 | #6 |
|-------|------|------|-------|-------|--------|------|
| CADJ | 0.63 | 5.63 | 0.65 | 0.51 | 0.40 | 0.92 |
| DADJ | 0.45 | 5.86 | 0.47 | 0.93 | 0.71 | 0.88 |
| CVM | 0.42 | 5.96 | 0.47 | 0.95 | 0.77 | 0.85 |
| FDM | 0.17 | 6.70 | 21.21 | -8.87 | -23.20 | 0.54 |

**Table 1: Evaluation of the full adjoint model with the ADJ (Adjoint) against the CVM (Complex Variable Method) sensitivities of the concentrations of an accumulation-mode aerosol species ASO4J (µg/ m$^3$) at Hour 24 with respect to the concentrations of a gas species SO$_2$ (ppmV) at Hour 23. The perturbation size for the CVM is 1.E-12 and the one for the FDM 10%.**