# Peer review of "A Multiphase CMAQ Version 5.0 Adjoint"

_Geoscientific Model Development, 2019_

## Referee Comment (RC1) · Anonymous Referee #1 · 10 Jan 2020

The authors present a description and evaluation of the implementation of an adjoint methodology into CMAQ version 5.0. This method is compatible with all the major components of the CMAQ model, which is a step forward from previously published implementations in recent versions of CMAQ that only included the implementation of the adjoint approach for inert aerosol species. The authors evaluate the adjoint implementation in each of the major modules of CMAQ which allows for better confidence in the approach and also provides useful information about which modules are best suited to an adjoint. This could guide future decisions about which particular model components (such as inorganic thermodynamics) to include as part of the core model. Components better suited for sensitivity analysis might be a higher priority in situations where multiple choices exist and perform similarly in terms of speed and skill.

The manuscript is generally well organized and written. The use of brute-force sensitivity and finite difference as an evaluation approach is novel. One concern is the

illustrative example at the end. It is very helpful to have an illustrative example of the type of information the adjoint provides, but the Figures (Figure 14) related to the illustrative example are confusing to interpret. The Figure caption suggests annual monetized health benefits normalized by emissions are presented. However, it is not clear whether the monetized benefits are normalized by national emissions or emissions from that same grid cell. Further, it is confusing to think about monetized health effects in places where no people reside (over the ocean for instance) and also where there are little to no emissions (northern Ontario near Hudson Bay). Perhaps there is a alternative illustration of the type of information the adjoint provides which would be simpler to interpret—such as looking at concentrations relative to some source/region and not even get into converting the concentrations to health effects.

---

## Referee Comment (RC2) · Anonymous Referee #2 · 19 Feb 2020

Summary

This is a very nice model development paper that summarizes a new capability in CMAQ with a potentially wide range of future applications for source attribution, inverse modeling, etc. Having both discrete and continuous adjoints as necessary for different processes along with the use of FDM vs CVM when necessary makes this novel for an air quality model like CMAQ. The authors have systematically broken down the CMAQ model into each of its major atmospheric processes and discussed both the implementation and evaluation of the adjoint technique, along with a policy-relevant illustration at the end.

Major Comments:

Given the motivation for this development to go beyond the earlier version of CMAQ Adjoint for gas-phase chemistry and demonstrate capability to model PM2.5, I find it

extremely limiting that the evaluation scenario used only a 7-day model simulation, and all evaluation is apparently shown only for a single hour (last hour of a day). While I appreciate the resource requirements for a longer time period, with the growth in computing technologies, it would have been valuable if the evaluation was performed for a one-month period at the minimum to ensure that the results are robust. Also, showing the evaluation for a 24-hour average (in addition to the single hour shown mostly) would also be policy-relevant given the short-term form of the health-based standard for PM2.5.

Further, the species used for evaluation is very selected (mostly ASO4J) and not robust and comprehensive. I suggest the authors quantify the evaluation metrics for all major PM2.5 constituents for each process when applicable.

Specific Comments:

Page 2 Line 26: Please add full citation for Constantin and Barrett, 2014 to the list of references. It is missing now.

Page 4 Line 5: Can the authors add a brief description of how the CMAQ adjoint capabilities as described in this study are different from the GEOS-Chem adjoint in Henze et al, 2007 or any other updates since then?

Page 7 Line 6: Should deposition (dry and wet) be added to the list of science processes in CMAQ? I do see later on that the authors justify not developing an adjoint for the deposition process.

Page 10 Section 2.5. This section discusses development of a python-based adjoint forcing pre-processor and mentions ability to calculate local maximum 8-hr average O3 concentration. However, the final policy illustration in Section 4 uses a PM2.5 case study. Please clarify if the python-based pre-processor was enhanced for this application as well, or if a different approach was used for the demonstration case.

Page 11 Section 3: For details about the other inputs used in this study, the reference

is to Turner et al (2015a). But that study used 12km modeling, while this is using 36km. Please clarify and reconcile this apparent discrepancy.

Page 12 Section 3.2.1. Why was AALKJ chosen here for the evaluation of the aerosol module? Can you provide some justification?

Page 13 Line 17: Can you quantify this "acceptable accuracy", or provide a reference?

Page 16 Line 1: If I understand this right, the adjoint agrees with the CVM, with a relative error less than 10%. Isn't this rather high to be acceptable? If so, for an emissions sector that has a 5% contribution, the results are within the error that the adjoint model produces, and hence cannot be meaningfully used for that range? Please provide some context to this 10% so that future users of this technique do not misuse it, and are aware of the limitations. In fact, I suggest that a separate section of limitations be added to point out other such issues.

Page 16 Line 3: "The problem with the FDM has been discussed earlier" Please provide exact Section where it was discussed earlier

Page 16 Section 3.4: Can you provide this information in a table? It may be easier on the reader

Page 17: For the model application case, even though it is illustrative, please provide additional information on the source of meteorology and emissions inputs that were used.

Page 17 Line 3: Change "IO" to "I/O"

Page 17: Line 19: "extrapolating from April to the full year". Should this really be "7 days in April" as stated in Section 3?

Page 17 Line 24: Why NO and not NOx (NO + NO2)?

Page 18 Line 18: The language re code availability seems to indicate that the model is not ready for further dissemination given the need for expanded user testing. Please

clarify.

Figures 1- 8: When you say "final concentration", is the spatial plot for the last hour of one-day? Did that one day have a spinup or is truly the first day of the modeling for this development? Is that same one hour used in the scatter plot, or all hours from the one day? Suggest that all hours for a single day be used in the scatter.

Figure 9: Which grid-cell was used in showing this Jacobian? Can you provide some context for the choice of this grid-cell and if it is representative for the whole domain? It will be helpful to see how these differences propagate through the 7-day period (or at least a month if feasible) that was modeled. Perhaps, show that as a time-series?

Figure 7 Caption: Change "The perturbation size for the FDM is 0.01/0.001/0.0001 ppb for the three figures" to clarify which figure has what size – top to bottom or bottom to top

Figure 7: Can you explain the non-zero values for FDM ranging between 0 – 3000 when ADJ = 0, for all top 3 figures?

Figure 11: What is the impact of this numerical noise in layers aloft in the adjoint of the transport scheme, and how would this affect the model results? I see a note on page 15 Lines 1-2 that "these are not uncommon and desirable fix does not appear possible."

Figure 11 Caption: There is really no "top plots" here. Please reword.

Figure 14 Caption: Should "long-term PM2.5 exposure" really be "7-day PM2.5 exposure"?

Table 1: Please define acronyms such as CADJ, DADJ, etc. What are cells 1 – 6 in this table? There is a lot of information in this table for evaluating the full adjoint, but the discussion of this table is very skinny. For e.g., look at cells #3 and #5. The FDM results range so widely (from 21.21 to -23.2), while both CADJ and DADJ results are much closer. What does this mean?

---

## Author Comment (AC1) · 18 Mar 2020

COMMENT: The authors present a description and evaluation of the implementation of an adjoint methodology into CMAQ version 5.0. This method is compatible with all the major components of the CMAQ model, which is a step forward from previously published implementations in recent versions of CMAQ that only included the implementation of the adjoint approach for inert aerosol species. The authors evaluate the adjoint implementation in each of the major modules of CMAQ which allows for better confidence in the approach and also provides useful information about which modules are best suited to an adjoint. This could guide future decisions about which particular model components (such as inorganic thermodynamics) to include as part of the core model. Components better suited for sensitivity analysis might be a higher priority in situations where multiple choices exist and perform similarly in terms of speed and skill.

[Figure]

The manuscript is generally well organized and written. The use of brute-force sensitivity and finite difference as an evaluation approach is novel. One concern is the illustrative example at the end. It is very helpful to have an illustrative example of the type of information the adjoint provides, but the Figures (Figure 14) related to the illustrative example are confusing to interpret. The Figure caption suggests annual monetized health benefits normalized by emissions are presented. However, it is not clear whether the monetized benefits are normalized by national emissions or emissions from that same grid cell. Further, it is confusing to think about monetized health effects in places where no people reside (over the ocean for instance) and also where there are little to no emissions (northern Ontario near Hudson Bay). Perhaps there is a alternative illustration of the type of information the adjoint provides which would be simpler to interpret such as looking at concentrations relative to some source/region and not even get into converting the concentrations to health effects.

RESPONSE: We appreciate the reviewer's comment and concern about the clarity of the illustrative example. While we agree with the reviewer that other examples may be more intuitive and easier to follow, we believe that source attribution of health impacts as location-specific BPTs is one of the most relevant, lucid, and practically significant examples of unique capabilities that the adjoint approach offers in the area of policy analysis. To address the reviewer's concern we have completely revised the section, to better explain the process for adjoint-based source attribution of health impacts, and the meaning of the calculated BPTs.

The reviewer is correct that locations with no population or emissions can have large BPTs. We define the adjoint cost function as the benefit over the entire US domain. What we obtain from the adjoint simulation (with unit conversion in post-processing) is the location-specific BPTs. In other words, the BPT values shown in Figure 14 are not normalized by emissions and only suggest how much benefit we would gain (or how much damage we would cause) if we cut a ton (or add a ton) of emissions of a pollutant at a specific location. As benefits are considered at the national scale (i.e.,

for the entire contiguous U.S.), emissions at locations with no population or emission could have an impact on health due to transport. As part of revising this section, and to address this specific point, the following is added to the manuscript"

"While the adjoint cost function is defined based on PM2.5 long-term mortality in the US alone, location-specific BPTs also provide a measure of cross-border impact. Finally, we note that BPTs are measures of marginal rather than total societal impact across the U.S., and as such, even areas with little or no emissions may show sizeable BPT estimates."

Finally, the reviewer's points about different science modules and their performance with respect to formal sensitivity analysis are well taken. While we agree with these comments, we would also like to point out that evaluation of numerical approaches and algorithms based on their performance in sensitivity analyses and their differentiability is a new and emerging concept in air quality and atmospheric modeling. Historically, these models have not been developed with differentiability in mind, but with accuracy and computational efficiency as the main drivers. As a result of ensuing practical trade-offs, discontinuities abound throughout CMAQ, as well as in other CTMs. These discontinuities are encountered in most science modules such as (in addition to inorganic thermodynamics) cloud processes, advection, mode-merging, SOA formation, native solvers of gaseous and aqueous chemistry, etc. We believe it will be a gradual but continued effort among the modeling community to address the issue of differentiability in future generations of algorithms used in CTMs.

To emphasize this important point we have added the following to the manuscript (section 3.2.1):

"The example given above is one of numerous cases where FDM was found to be inaccurate or inadequate in evaluating adjoint sensitivities. The inadequacy of FDM in producing accurate sensitivity estimates is due to process nonlinearities, as well as discontinuities that exist throughout CMAQ. This is the case in a number of CMAQ

processes such as SOA formation, inorganic thermodynamics, clouds, aqueous chemistry, advection, etc. This issue is not limited to CMAQ alone and exists in all air quality models, as providing a smooth solution for the governing equations may be lost in trade-offs for added computational efficiency, improving stability, or reducing numerical artifacts in the development stage."

And the following to our conclusion:

"We find that the development of adjoint versions of air quality and atmospheric models is often complicated by the abundance of discontinuities throughout these models that make differentiation challenging. Historically, these models have not been developed with differentiability in mind, but with accuracy and computational efficiency as the main drivers. As the development and applications of formal sensitivity analysis tools (such as adjoint models) become more prevalent, there is a need for a gradual but sustained effort by the modeling community to consider differentiability as an additional design constraint in future developments."

---

## Author Comment (AC2) · 18 Mar 2020

Summary

This is a very nice model development paper that summarizes a new capability in CMAQ with a potentially wide range of future applications for source attribution, inverse modeling, etc. Having both discrete and continuous adjoints as necessary for different processes along with the use of FDM vs CVM when necessary makes this novel for an air quality model like CMAQ. The authors have systematically broken down the CMAQ model into each of its major atmospheric processes and discussed both the implementation and evaluation of the adjoint technique, along with a policy-relevant illustration at the end.

COMMENT: Given the motivation for this development to go beyond the earlier version of CMAQ Adjoint for gas-phase chemistry and demonstrate capability to model

PM2.5, I find it extremely limiting that the evaluation scenario used only a 7-day model simulation, and all evaluation is apparently shown only for a single hour (last hour of a day). While I appreciate the resource requirements for a longer time period, with the growth in computing technologies, it would have been valuable if the evaluation was performed for a one-month period at the minimum to ensure that the results are robust. Also, showing the evaluation for a 24-hour average (in addition to the single hour shown mostly) would also be policy-relevant given the short-term form of the health-based standard for PM2.5.

RESPONSE: We thank the reviewer for pointing out a source of confusion in the manuscript. All evaluations are done for a full day. However, the adjoint model is only forced at the last time-step, and the gradient is then evaluated after 24 hours of backward simulations. While using continuous forcing, for instance one that corresponds to 24-hour average concentration would be more policy-relevant, we have instead used pulse, instantaneous forcing. The choice of pulse forcing for the adjoint evaluation has two reasons. First, this choice allows for a more straightforward testing process (for example, the calculation of finite difference or CVM sensitivities), especially when a large number of tests need to be conducted. Second, using instantaneous forcing provides a more stringent evaluation framework for the adjoint results. The magnitude of the impact of a source often diminishes gradually, or at times precipitously, with time and distance. This means that receptors that are closer to a source, both horizontally and vertically, are likely to be impacted by the source at a higher rate. In adjoint sense, this means that nearby sources are likely to show more prominently when those receptors are forced. The same is true about evolution of influences in time for processes that do not include transport and are only integrated in time; time steps that are closer to the initial forcing, would have the largest influence. As a result, if forcing is done continuously, then the adjoint estimate is dominated by "local" (in time and space) influences. Under continuous forcing, and with this large "local" component, it would become more difficult to evaluate (for example in comparison with brute-force) how the adjoint influences are sustained over longer times or larger distances. We believe that

a daylong simulation (translating to an average transport range of few hundred kilometers) with pulse forcing, as implemented, would provide a signal for the evaluation that has sufficiently evolved but is not overly diminished.

The reviewer's point about the need for a longer simulation period in our application example, i.e. source attribution of health impacts, is well taken. We have extended our simulation time to two full warm and cold seasons (see revised Figure 14).

COMMENT: Further, the species used for evaluation is very selected (mostly ASO4J) and not robust and comprehensive. I suggest the authors quantify the evaluation metrics for all major PM2.5 constituents for each process when applicable.

RESPONSE: As stated in the manuscript (original, Page 12 Line 31; revised, Page 13 Line 8), our intention is to make, within computational constraints, the evaluation framework more comprehensive in terms of physical pathways and numerical procedures. Testing all source-receptor species pairs is not feasible, and summing up the PM species would obfuscate important processes. We have chosen the source-receptor species pair to best match the process being evaluated. For example, we use AALKJ for the secondary organic aerosol process, ASO4J for the aerosol thermodynamics module ISORROPIA/ANISORROPIA, and O3 for chemistry. For transport processes, the choice does not matter; we use ASO4J, for consistency.

The above-mentioned species are those chosen as receptor species; there are other species used as sources. For example, we use ASO4J as both receptor and source for clouds dynamics, but SO2 is adopted as the source to account for the impact through aqueous chemistry on ASO4J. For aerosols, we choose ASO4J as receptor and ASO4I as source to test intermodal (e.g., coagulation or mode-merging) processes. In cases where we have encountered outstanding discrepancies, we have used the Jacobian of sensitivities (i.e., all possible source-receptor species pairs) related to a process to investigate possible causes, as done for the chemistry process.

Specific Comments:

COMMENT: Page 2 Line 26: Please add full citation for Constantin and Barrett, 2014 to the list of references. It is missing now.

RESPONSE: Added: Constantin, B. V. and Barrett, S. R.: Application of the complex step method to chemistry-transport modeling. Atmos. Environ., 99, 457-465, 10.1016/j.atmosenv.2014.10.017, 2014.

COMMENT: Page 4 Line 5: Can the authors add a brief description of how the CMAQ adjoint capabilities as described in this study are different from the GEOS-Chem adjoint in Henze et al, 2007 or any other updates since then?

RESPONSE: The difference between the two adjoint models are inherited from the two primal models, CMAQ and GEOS-Chem. GEOS-Chem is a global model with coarser resolution, compared to the regional CMAQ model. As a global model, development efforts for GEOS-Chem have had more of a focus (compared to CMAQ) on processes that impact long-range transport (convective mixing, stratospheric intrusion, etc), long-lived species (e.g., CO, methane), and global budgets of atmospheric constituents (e.g., lightning emissions). On the other hand, CMAQ, as a limited area model has seen more development and details in processes that affect surface concentrations. Of course this distinction is rather simplistic, and the scales of the two models have approached in past years, as CMAQ now has a hemispheric version, and GEOS-Chem is capable of higher-resolution simulations in nested configurations. These differences in the base models are also reflected in the adjoint versions.

We do not include a detailed discussion of differences between GEOS-Chem and CMAQ adjoints in the manuscript, as we feel the scale and applications of the two models are quite different such that a comparison would be beyond the scope of a manuscript about CMAQ. However the manuscript is revised to read :" Adjoint of air quality models . . ., or were developed for a global model with coarser resolution and varying levels of detail in representation of some of the atmospheric processes (Henze et al., 2007)."

COMMENT: Page 7 Line 6: Should deposition (dry and wet) be added to the list of science processes in CMAQ? I do see later on that the authors justify not developing an adjoint for the deposition process.

RESPONSE: We follow the convention of CMAQ about the science processes. The wet deposition is dealt with in clouds and the dry deposition is part of the vertical diffusion science process. The science processes are first mentioned in the beginning of Section 2. Both wet and dry deposition are included in the development; however, the adjoint for bidirectional deposition of ammonia is not available, and as such ammonia has conventional dry deposition in the current version.

A reference and more details are added to clarify: "As mentioned at the beginning of Section 2, the science processes in CMAQ include advection, horizontal and vertical diffusion (including dry deposition), gas-phase chemistry, aerosols (including thermo-dynamics and dynamics), and clouds (including aqueous chemistry and wet deposition)."

We also correct the following reference in the "Conclusions" section:

"Some components of CMAQ that do not yet have an adjoint include the bidirectional dry deposition in vertical diffusion and photolysis rate calculations in gas-phase chemistry."

COMMENT: Page 10 Section 2.5. This section discusses development of a python-based adjoint forcing pre-processor and mentions ability to calculate local maximum 8-hr average O3 concentration. However, the final policy illustration in Section 4 uses a PM2.5 case study. Please clarify if the python-based pre-processor was enhanced for this application as well, or if a different approach was used for the demonstration case.

RESPONSE: For the PM2.5 application, we used Fortran instead of Python. However, extending the Python tool to include PM2.5 is planned.

COMMENT: Page 11 Section 3: For details about the other inputs used in this study,

the reference is to Turner et al (2015a). But that study used 12km modeling, while this is using 36km. Please clarify and reconcile this apparent discrepancy.

RESPONSE: The CMAQ inputs including meteorology, emissions, initial and boundary conditions are prepared using the same configuration, except for the difference in resolution. The manuscript has been revised to clarify:

"More details about meteorological inputs, initial and boundary conditions, and emissions are provided in Turner et al. (2015a), where a version of the dataset with12-km horizontal resolution was used."

COMMENT: Page 12 Section 3.2.1. Why was AALKJ chosen here for the evaluation of the aerosol module? Can you provide some justification?

RESPONSE: AALKJ is chosen as a relevant example, in this case for the SOA formation process. In this particular case, among various semi-volatile species, AALKJ was chosen because it had poor agreement against FDM, but performed very well in comparison to the CVM. As stated in the manuscript:

"For this process, the FDM behaved well for most of the test cases (results not shown); in the few cases when the ADJ and FDM did not agree and tuning with the perturbation sizes did not help, use of CVM demonstrated good accuracy of adjoint results (i.e., agreement along one-on-one line). This is an example of numerous cases where FDM was found to be inaccurate or inadequate in evaluating adjoint sensitivities."

COMMENT: Page 13 Line 17: Can you quantify this "acceptable accuracy", or provide a reference?

RESPONSE: Revised as (Page 13 Line 30 in the revised manuscript): "Overall, our testing confirms the findings in Capps et al. (2012) that the CVM implementation of ISORROPIA produces approximations that agree with the adjoint results."

COMMENT: Page 16 Line 1: If I understand this right, the adjoint agrees with the CVM, with a relative error less than 10%. Isn't this rather high to be acceptable? If so, for

Interactive
comment

an emissions sector that has a 5% contribution, the results are within the error that the adjoint model produces, and hence cannot be meaningfully used for that range? Please provide some context to this 10% so that future users of this technique do not misuse it, and are aware of the limitations. In fact, I suggest that a separate section of limitations be added to point out other such issues.

RESPONSE: The 10% difference mentioned refers to the agreement between CADJ and the CVM, as DADJ and CVM agree very well. As mentioned in the manuscript, the difference between CADJ and CVM should not be taken as inaccuracy of CADJ, but as a manifestation of different approaches in estimating source-receptor relationships. This point is discussed in the manuscript:

"Results shown in Table 1 suggest that the discrete adjoint has a better agreement with CVM than the continuous adjoint. However, it is important to note that better agreement between the discrete adjoint and CVM should not be understood as better accuracy of the discrete adjoint in comparison with continuous adjoint. The numerical solution to the advection equation entails inherent truncation errors from discretization schemes. These errors exist in solving the forward or adjoint advection equations; however the discrete adjoint by design remains loyal to, and consistent with the errors in the forward application (CVM in this case), while numerical solution to the continuous adjoint will result in different and inconsistent errors. The continuous adjoint is a different representation of the impacts on the adjoint cost function, but of similar mathematical accuracy, when compared to the forward or tangent linear model; therefore, the numerical solution to it should be considered as accurate as the discrete adjoint, regardless of the agreement with forward-based benchmarks such as CVM."

The statement is also clarified to better reflect differences between CADJ, DADJ, and CVM:

"In general, the adjoint models, particularly DADJ, agree well with the CVM, while in the case of CADJ a larger relative error exists in comparison with the CVM"

A reference to the above discussion is added to the caption of Table 1:

"Table 1: Evaluation of the full adjoint model with the CADJ (Continuous Adjoint)/DADJ (Discrete Adjoint) against the CVM (Complex Variable Method)/FDM (Finite Difference Method) sensitivities of the concentrations of an accumulation-mode aerosol species ASO4J ($\mu$g/ m3) at hour 24 with respect to the concentrations of a gas species SO2 (ppmV) at hour 23. The cells are arbitrarily picked. The perturbation size for the CVM is 1.E-12 and the one for the FDM 10%. The relation of FDM and CVM sensitivities with CADJ and DADJ results has been discussed in Section 3.3."

COMMENT: Page 16 Line 3: "The problem with the FDM has been discussed earlier" Please provide exact Section where it was discussed earlier

RESPONSE: Revised: "The problem with the FDM has been discussed at the beginning of Section 3 and is not repeated here. "

COMMENT: Page 16 Section 3.4: Can you provide this information in a table? It may be easier on the reader

RESPONSE: Table 2 added about checkpoint file size for each science process. Revised on p. 16: "A summary of the checkpointing file sizes is provided in Table 2."

"Table 2: Sizes of checkpoint files for the science processes in CMAQ for a single day. The computational domain has 148 columns, 112 rows, and 24 vertical layers. The synchronization time step of CMAQ is 12 minutes. Shown for horizontal and vertical advections are checkpointing file sizes from the continuous version of adjoint."

COMMENT: Page 17: For the model application case, even though it is illustrative, please provide additional information on the source of meteorology and emissions inputs that were used.

RESPONSE: Details added for the new results:" For the backward sensitivity analysis, we run the adjoint for the year of 2016 for the contiguous U.S. domain with 36-km resolution inputs from the Intermountain West Data Warehouse (National Emissions

Inventory Collaborative, 2019). The computational domain contains 172 columns and 148 rows with 35 vertical layers."

COMMENT: Page 17 Line 3: Change "IO" to "I/O"

RESPONSE: Revised as suggested.

COMMENT: Page 17: Line 19: "extrapolating from April to the full year". Should this really be "7 days in April" as stated in Section 3?

RESPONSE: We have now extended the results to two full 3-month seasons based on a different dataset.

COMMENT: Page 17 Line 24: Why NO and not NOx (NO + NO2)?

RESPONSE: NOx is shown for the new results.

COMMENT: Page 18 Line 18: The language re code availability seems to indicate that the model is not ready for further dissemination given the need for expanded user testing. Please clarify.

RESPONSE: We intend to publicly release the model later in 2020 and after it has gone through limited release and a wider variety of simulations (different domains, scales, etc).

COMMENT: Figures 1- 8: When you say "final concentration", is the spatial plot for the last hour of one-day? Did that one day have a spinup or is truly the first day of the modeling for this development? Is that same one hour used in the scatter plot, or all hours from the one day? Suggest that all hours for a single day be used in the scatter.

RESPONSE: Plotted in all the plots are the sensitivities of instantaneous concentration at Hour 24 (final) to instantaneous concentrations at Hour 0 (initial). To produce sensitivities to concentrations at each hour, it would require 23 extra runs for the FDM or CVM which is not feasible. As seen in the plots, simulations across the computational space provide sufficient points of comparison.

There is a 6-day spin-up period; the validation is performed on day 7. Revised on p.11:

"The air quality simulation scenario used for the evaluation is for the contiguous U.S. domain with a 36-km horizontal resolution and 24 vertical layers for the first seven days of April 2008, with the first 6 days used for spin-up."

COMMENT: Figure 9: Which grid-cell was used in showing this Jacobian? Can you provide some context for the choice of this grid-cell and if it is representative for the whole domain? It will be helpful to see how these differences propagate through the 7-day period (or at least a month if feasible) that was modeled. Perhaps, show that as a time-series?

RESPONSE: While the Jacobian includes interesting and valuable information, its creation is time-consuming and resource intensive. In the current scope of work, we have used the Jacobian as a diagnostic tool and only when we encountered problems that we were not able to resolve through conventional means and efforts. The cell for which the results are shown was chosen as it presented a problematic case, where our adjoint results did not agree with CVM, and therefore, we constructed the full Jacobian.

COMMENT: Figure 7 Caption: Change "The perturbation size for the FDM is 0.01/0.001/0.0001 ppb for the three figures" to clarify which figure has what size – top to bottom or bottom to top Figure 7: Can you explain the non-zero values for FDM ranging between 0 – 3000 when ADJ = 0, for all top 3 figures?

RESPONSE: Revised as: "The perturbation size for the FDM is 0.01/0.001/0.0001 ppb for the top three figures (from top to bottom)."

Thresholds are commonly used in CMAQ to make concentrations non-zero for computational purposes, say, to avoid division by zero. When such circumstances occur, the adjoint would give zero sensitivities but the finite difference with an absolute perturbation size would not. This is one of the instances of CMAQ (and other CTMs) having a non-smooth and discontinuous response surface that would not lend itself to accurate finite difference approximation. Considering model complexity, we use sensitivities normalized by the initial concentrations to check if there is an agreement between ADJ and FDM. And there is.

COMMENT: Figure 11: What is the impact of this numerical noise in layers aloft in the adjoint of the transport scheme, and how would this affect the model results? I see a note on page 15 Lines 1-2 that "these are not uncommon and desirable fix does not appear possible."

RESPONSE: In Figure 11, the y-axis in the tile plots represents the vertical layers. As can be seen, the numerical noise reaches out to all layers.

COMMENT: Figure 11 Caption: There is really no "top plots" here. Please reword.

RESPONSE: Revised as: "For the tile plots, the x- and y-axes represent the horizontal y direction and the vertical layers, respectively."

COMMENT: Figure 14 Caption: Should "long-term PM2.5 exposure" really be "7-day PM2.5 exposure"?

RESPONSE: We have now extended the results to two full 3-month seasons based on a different dataset.

COMMENT: Table 1: Please define acronyms such as CADJ, DADJ, etc. What are cells 1 – 6 in this table? There is a lot of information in this table for evaluating the full adjoint, but the discussion of this table is very skinny. For e.g., look at cells #3 and #5. The FDM results range so widely (from 21.21 to -23.2), while both CADJ and DADJ results are much closer. What does this mean?

RESPONSE: The acronyms are now added. The cells are picked in an arbitrary way, loosely based on the sensitivity values of the continuous adjoint (CADJ). The values from the FDM merely suggest that a 10% perturbation size is not able to accurately produce the sensitivity information, as encountered many times in the manuscript. This also points to inadequacy of FDM in estimating single-source impacts within a model

such as CMAQ that has a fragmented response surface. The Table caption is revised:

"Table 1: Evaluation of the full adjoint model with the CADJ (Continuous Adjoint)/DADJ (Discrete Adjoint) against the CVM (Complex Variable Method)/FDM (Finite Difference Method) sensitivities of the concentrations of an accumulation-mode aerosol species ASO4J ($\mu$g/ m3) at Hour 24 with respect to the concentrations of a gas species SO2 (ppmV) at Hour 23. The cells are arbitrarily picked. The perturbation size for the CVM is 1.E-12 and the one for the FDM 10%."

––––––––––––––––––––––––––––––

---

## Author Response (AR2)

Dear Dr. Topping,

We would like to thank you for your review of our manuscript and your comments on code availability. As mentioned in the original submission, the CMAQ-Adjoint code will be hosted by the U.S. EPA. However, EPA requires further testing on platform and compiler compatibility, which given the current limitations (i.e., working from home) can take 2-3 months. As a solution, we have included EPA's GitHub address as the permanent repository in the manuscript. However, while the testing at EPA is ongoing, this address would redirect users to a public Zenodo repository of the code. When EPA's testing is complete, the code will be hosted at the permanent address at EPA's GitHub. We hope that this solution is acceptable to the journal.

Best regards,

CMAQ Adjoint Development Team

[revised manuscript text omitted]